# Epidemiology of community-acquired pneumonia among hospitalised children in Indonesia: a multicentre, prospective study

Dewi Lokida,[1] Helmia Farida,[2] Rina Triasih,[3] Yan Mardian  ,[4] Herman Kosasih,[4] Adhella Menur Naysilla,[4] Arif Budiman,[1] Chakrawati Hayuningsih,[1] Moh Syarofil Anam,[2] Dwi Wastoro,[2] Mujahidah Mujahidah,[3] Setya Dipayana,[2] Amalia Setyati,[3] Abu Tholib Aman,[3] Nurhayati Lukman,[4] Muhammad Karyana,[5] Ahnika Kline,[6] Aaron Neal,[6] Chuen-Yen Lau,[7] Clifford Lane[6]

For numbered affiliations see end of article.

**Correspondence to**
Dr Yan Mardian;
ymardian@ina-respond.net

## ABSTRACT

**Objective** To identify aetiologies of childhood community-acquired pneumonia (CAP) based on a comprehensive diagnostic approach.

**Design** 'Partnerships for Enhanced Engagement in Research-Pneumonia in Paediatrics (PEER-PePPeS)' study was an observational prospective cohort study conducted from July 2017 to September 2019.

**Setting** Government referral teaching hospitals and satellite sites in three cities in Indonesia: Semarang, Yogyakarta and Tangerang.

**Participants** Hospitalised children aged 2–59 months who met the criteria for pneumonia were eligible. Children were excluded if they had been hospitalised for >24 hours; had malignancy or history of malignancy; a history of long-term (>2 months) steroid therapy, or conditions that might interfere with compliance with study procedures.

**Main outcome(s) measure(s)** Causative bacterial, viral or mixed pathogen(s) for pneumonia were determined using microbiological, molecular and serological tests from routinely collected specimens (blood, sputum and nasopharyngeal swabs). We applied a previously published algorithm (PEER-PePPeS rules) to determine the causative pathogen(s).

**Results** 188 subjects were enrolled. Based on our algorithm, 48 (25.5%) had a bacterial infection, 31 (16.5%) had a viral infection, 76 (40.4%) had mixed bacterial and viral infections, and 33 (17.6%) were unable to be classified. The five most common causative pathogens identified were *Haemophilus influenzae* non-type B (N=73, 38.8%), respiratory syncytial virus (RSV) (N=51, 27.1%), *Klebsiella pneumoniae* (N=43, 22.9%), *Streptococcus pneumoniae* (N=29, 15.4%) and Influenza virus (N=25, 13.3%). RSV and influenza virus diagnoses were highly associated with Indonesia's rainy season (November–March). The PCR assays on induced sputum (IS) specimens captured most of the pathogens identified in this study.

**Conclusions** Our study found that *H. influenzae* non-type B and RSV were the most frequently identified pathogens causing hospitalised CAP among Indonesian children aged 2–59 months old. Our study also highlights the importance

## STRENGTHS AND LIMITATIONS OF THIS STUDY

⇒ Prospective multisite study conducted over 27 months.
⇒ Used a comprehensive diagnostic approach (culture, molecular testing and paired serological assays) to identify causative pathogens from routinely collected specimens (blood, sputum and nasopharyngeal swabs).
⇒ The relatively small sample size, geographical limitation to the island of Java and observational design limit generalisability and causal inference.
⇒ We did not collect lung aspirates or pleural fluid specimens, which are preferred for determination of pneumonia aetiology, and did not include healthy control children, limiting ability to estimate the adjusted population attributable fraction for each pathogen.
⇒ Several cases of pneumonia were attributed to unknown aetiology, which could be due to administration of antibiotics before culture, poor sputum quality, limited bacterial and viral panels, lack of fungal testing or another factor.

of PCR for diagnosis and by extension, appropriate use of antimicrobials.

**Trail registration number** NCT03366454

## INTRODUCTION

Pneumonia is the leading infectious cause of child mortality, with a greater burden in low-income and middle-income countries (LMICs).[1] In Indonesia, pneumonia contributed to 15% of childhood deaths and was the second leading cause of death among children under 5 years in 2017.[2] Indonesian practice guidelines are adapted from the WHO guidelines, which are based on 1970s–1990's data showing bacteria such as *Haemophilus influenzae* type b (Hib) and *Streptococcus*

*pneumoniae* caused the majority of fatal pneumonias in children.[3-5] Therefore, empiric antibiotics are considered first-line treatment for children with community-acquired pneumonia (CAP).[6-8] Despite evidence that appropriate antibiotics are lifesaving, rational selection of antibiotics for pneumonia is hampered by low adherence to guidelines and scarcity of point-of-care diagnostics.[9-11] Consequently, healthcare providers, particularly those in LMIC, are likely to overtreat non-bacterial pneumonia with antibiotics.[11 12]

Several recent studies of CAP in children have highlighted the role of viral aetiologies. Increased recognition of viral aetiologies of CAP is likely due to both enhanced molecular diagnostic capacity and wide deployment of Hib and pneumococcal conjugate vaccines (PCV).[13 14] Treatment of non-bacterial pneumonia with antibiotics may engender avoidable antimicrobial resistance. Thus, current data on the aetiologies of childhood pneumonia is needed and should be regularly evaluated to inform vaccination policies, empiric management decisions and targeted treatment.[12]

From a diagnostic standpoint, direct demonstration of organisms by culture (or staining) of lung aspirates has been the standard for determining microbial aetiology of CAP.[15] In the current era, many use less-invasive biological specimens (eg, blood, naso/oropharyngeal secretions, bronchoalveolar lavage or induced sputum (IS)) and employ diverse methods (eg, culture, PCR, antigen detection or paired serology) to identify organisms.[16] However, such comprehensive methods are costly and often require specialised equipment and human resources, limiting feasibility in low-resource settings.[17 18]

Prospective community-based cohort studies that define pathogen(s) causing CAP in Indonesian children are scarce. We conducted a 'Partnerships for Enhanced Engagement in Research-Pneumonia in Paediatrics (PEER-PePPeS)' study, which aimed to identify aetiologies of childhood CAP using comprehensive diagnostic methods.

## METHODS
### Study design and study sites
PEER-PePPeS was a multisite observational cohort study (ClinicalTrials.gov Identifier: NCT03366454) seeking to determine aetiologies of CAP among children aged 2–59 months in Indonesia. The study was conducted by the Indonesia Research Partnership on Infectious Disease (INA-RESPOND) and enrolled participants initially at three government referral teaching hospitals in three provinces: Kariadi Hospital (Central Java), Sardjito Hospital (Yogyakarta) and Tangerang District Hospital (Banten), as shown in online supplemental file 1). Satellite sites located near the primary sites were added during the study to facilitate subject recruitment.

### Study definitions
In this study, pneumonia in children was defined as cough or fever with at least one of the following: shortness of breath (indicated by at least one of the following signs: head bobbing; nasal flaring; chest indrawing or intercostal retracting), tachypnea, grunting, crackles, rhonchi, decreased vesicular breath sounds, bronchial breath sounds or chest X-ray findings consistent with pneumonia. Tachypnoea was defined as respiratory rate >50/min for infants 2–12 months and >40/min for children>12–60 months.[19] Abnormal chest X-ray findings consistent with pneumonia were defined as presence of either focal or diffuse infiltrates, a silhouette sign, pleural effusion or air bronchogram.[20] Chest X-rays were read by the paediatrician.

Based on WHO classification and treatment of childhood pneumonia at health facilities (2014 version), for children 2–59 months of age, severe pneumonia is defined as pneumonia (tachypnea and/or chest indrawing) accompanied by presence of any danger signs, including inability to drink, persistent vomiting, convulsions, lethargy or loss of consciousness, stridor in a calm child or severe malnutrition.[19]

### Study participants
PEER-PePPeS study enrolled children aged 2–59 months, who were hospitalised between 18 July 2017 and 25 September 2019, and met the definition for pneumonia. Eligible subjects were enrolled within 24 hours of admission. Children were excluded if they had been hospitalised for >24 hours; had a malignancy or history of malignancy; a history of long term (≥2 months) steroid therapy; or conditions that might interfere with compliance with study procedures (eg, very ill patients for whom specimens could not be obtained or living outside the area for which follow-up was practical).

### Study procedures
Demographic and anthropometric data, current signs and symptoms, pregnancy history, vaccination status, breastfeeding history, antibiotic and steroid exposure, family history, medical history, risk factors, haematology profiles, chemistry results and chest X-ray (per standard of care) were collected at enrolment. Clinical examination (vital signs, general examination, lung auscultation, SpO2); nasopharyngeal (NP) swab for molecular tests; IS for culture and molecular tests; collection of blood specimens for routine blood count, cultures, molecular tests, serological tests, C reactive protein (CRP) and procalcitonin (PCT) were also performed. We prospectively followed subjects daily until hospital discharge; data on vital signs, respiratory signs, intensive care admission, intubation, complications and treatment were collected. On day 14, we performed clinical examinations and collected convalescent sera for serology tests; subjects discharged before day 14 returned to clinic for their evaluation. We conducted a telephone interview on day 30 (±4 days) to assess clinical outcome.

This study used several widely available bacterial and viral respiratory molecular pathogen panels and serological assays.[21-24] NP and IS specimens were tested with a

PCR panel that included 12 viruses (influenza A, influenza B, adenovirus, enterovirus, bocavirus, respiratory syncytial virus (RSV) A, RSV B, human metapneumovirus (hMPV), rhinovirus, parainfluenza virus (PIV) 1–4, coronavirus OC43 and coronavirus NL63). NP specimens were evaluated by PCR for five bacteria (*Haemophilus influenzae, Streptococcus pneumoniae, Moraxella catarrhalis, Staphylococcus aureus* and *Klebsiella pneumoniae*), while IS specimens were tested for nine (*Haemophilus influenzae, Streptococcus pneumoniae, Mycoplasma pneumoniae, Chlamydia pneumoniae, Bordetella pertussis, Moraxella catarrhalis, Staphylococcus aureus, Klebsiella pneumoniae* and *Legionella pneumoniae*). Good quality (<10 squamous epithelial cells per low power field[12]) IS specimens underwent culture and gram stain.[25] For whole blood, qPCR was performed for three bacteria (*Haemophilus influenzae, Streptococcus pneumoniae* and *Staphylococcus aureus*). Serological testing for seven viruses (influenza A, influenza B, adenovirus, parvovirus B19, echovirus/enterovirus, RSV, PIV) and four bacteria (*Mycoplasma pneumoniae, Chlamydia pneumoniae, Legionella pneumoniae* and *Bordetella pertussis*) was performed using paired acute-convalescent sera.

Blood culture, IS culture and Gram stain, routine blood count, CRP, PCT and chest X-ray were performed by the laboratory/radiology department at the hospital site. qPCR and serology assays were performed retrospectively at the INA-RESPOND Reference Laboratory located in Tangerang District Hospital. Details of blood culture, sputum culture, molecular and serology test techniques are shown in online supplemental table 1.

### Pathogen identification

Causative bacterial, viral or mixed pathogen for the pneumonia was determined based on an algorithm (PEER-PePPeS rules) for interpretation of microbiological, molecular and serological test results published previously.[12] In brief, we considered all organisms detected by blood culture, detected by whole blood PCR, or that grew from good quality IS specimen in high quantities with a compatible primary Gram stain as potential causative bacterial pathogens. Bacteria commonly considered contaminants were excluded. For the nasopharynx, potential colonising bacteria (eg, *H. influenzae, S. pneumoniae* and *S. aureus*) and potential innocent bystander viruses (eg, bocavirus, adenovirus, non-SARS human CoVs (hCoVs), enterovirus and rhinovirus) were determined to be causative based on a PCR density cut-off and/or serodiagnosis criteria for paired acute and convalescent sera (seroconversion or a two to four-fold increase in antibody titers in the convalescent specimen).[12]

### Data collection and statistical analysis

Data were recorded on paper case report forms and entered in duplicate into OpenClinica (OpenClinica, Massachusetts, USA) by research staff. Categorical variables were summarised using absolute values and percentages, and continuous variables as medians and IQRs. Differences in categorical variables were compared using Pearson $\chi^2$ or Fisher's exact test when the expected values in any of the contingency table cells were below 5. Differences in continuous variables were compared using one-way analysis of variance or Kruskal-Wallis H-test for data which did not follow the normal distribution based on Levene's test. Statistical analyses were performed using SPSS software V.23 (IBM). All p values were two sided. Level of significance was set at $p<0.05$.

### Patient and public involvement statement

Patients or the public were not involved in study design or study conduct at any stage from inception to completion and dissemination of this project. Patients who met the eligibility criteria as described above were recruited to this study.

## RESULTS

### Study population

Of 444 children who were hospitalised with CAP, 188 (42.3%) were eligible and enrolled in the study. Of 256 screening failures, 31.8% were due to hospitalisation >24 hours at the time of screening and 22.1% to circumstances that might interfere with the study procedures. Of the 188 enrolled children, 184 (97.9%) had radiologic evidence of pneumonia. 179 (95.1%) subjects completed the study, including 19 (10.1%) who died. Eight subjects (4.3%) were lost to follow-up, and one subject (0.5%) withdrew from the study. The study flow is shown in figure 1.

Demographic and clinical characteristics are presented in table 1. Age, gender, laboratory values and pneumonia severity by WHO classification were similar across the three study sites. The median age was 9 months (IQR 5–20), and 54.7% of subjects were male. The most common comorbid conditions/medical histories were developmental delay (27.7%), congenital heart disease (26.1%), low birth weight (24.4%) and severe malnutrition (18.6%), with subjects from Yogyakarta site having the greatest proportion of those comorbidities. The percentage of subjects who had been vaccinated (age adjusted) against pneumococcus, influenza, Hib-DPT and measles vaccines were 2.1%, 1.1%, 55.9% and 75.0%, respectively.

The most common symptoms were shortness of breath (92.6%), cough (91.0%) and fever (80.9%). Signs noted during the initial examination included intercostal retraction (91.0%), rhonchi (89.4%) and chest indrawing (66.5%). Of 188 subjects, 172 (91.4%) and 167 (88.8%) had CRP and PCT measured with median values of 9.0 (IQR 3.6–28.0; Ref range ≤5) mg/L and 0.2 (IQR 0.1–1.7; Ref range ≤0.15) ng/mL, respectively. Interstitial infiltrate (69.7%) was the most common radiographic finding. 47.3% of cases were classified as severe pneumonia according to the WHO classification system. All 188 enrolled cases were treated with antibiotics, and 150 of them (79.8%) had

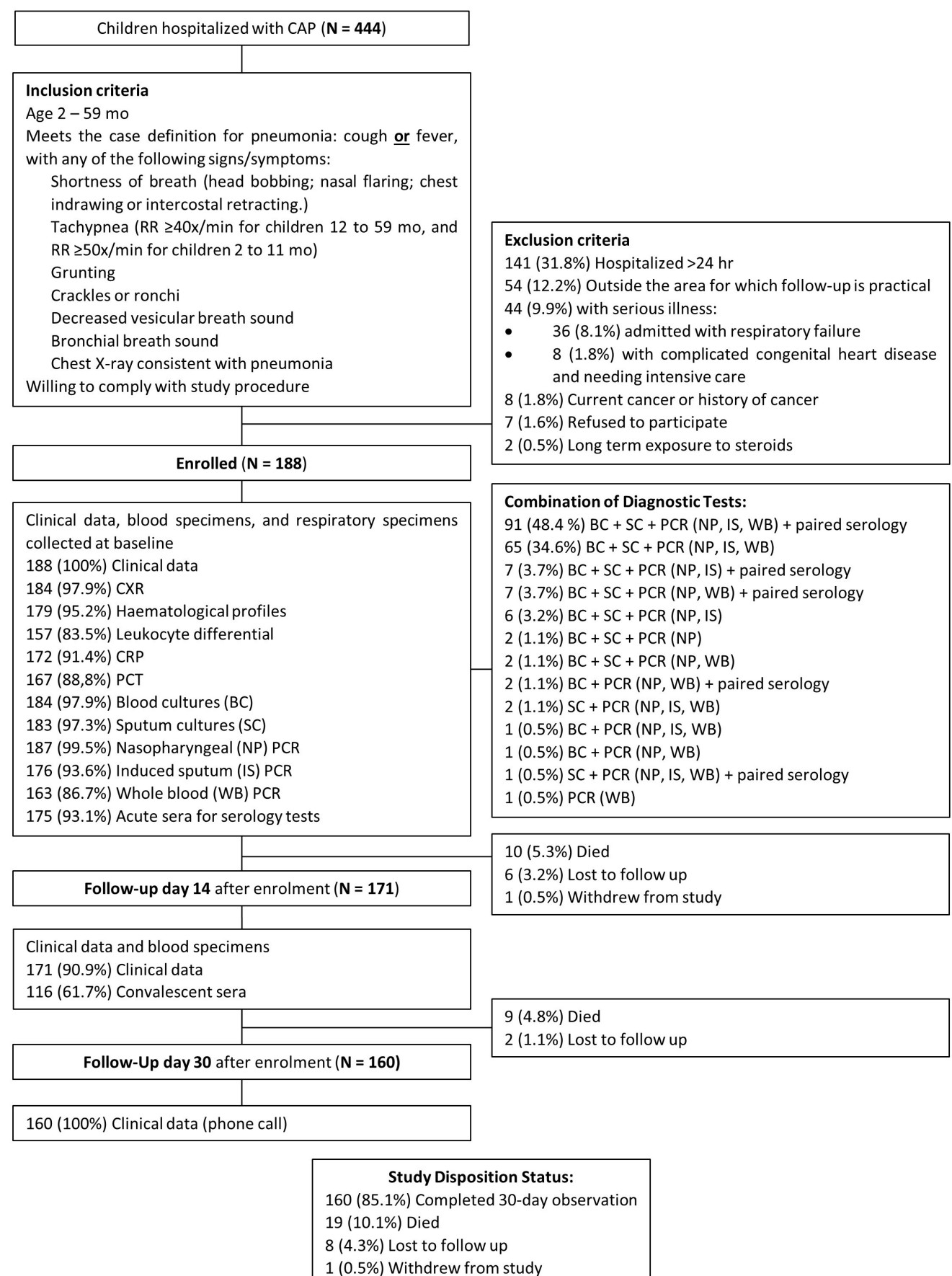

**Figure 1** Subject screening, enrolment and monitoring flow chart. CAP, community-acquired pneumonia; CRP, C reactive protein; CXR, chest X-ray; PCT, procalcitonin; RR, respiratory rate.

**Table 1** Baseline characteristics of subjects

| Demographic characteristics | All (N=188) | Semarang (N=47) | Yogyakarta (N=52) | Tangerang (N=89) | P value |
|---|---|---|---|---|---|
| Age, median (IQR) months | 9 (5–20) | 9 (5.5–21) | 8 (4–13.3) | 11 (5–20) | 0.442 |
| Gender, male, (%) | 103 (54.7) | 29 (61.7) | 26 (50) | 48 (53.9) | 0.493 |
| Household characteristics, (%) | | | | | |
| Low education of parents* | 163 (86.7) | 37 (78.7) | 43 (82.7) | 84 (94.3) | 0.019 |
| Living in a dense neighbourhood† | 121 (64.4) | 19 (40.4) | 42 (80.8) | 60 (67.4) | <0.001 |
| Exposure to cigarette smoke | 120 (63.8) | 24 (51.1) | 27 (51.9) | 69 (77.5) | 0.001 |
| Sick household contact <14 days | 109 (58.0) | 22 (46.8) | 43 (82.7) | 44 (49.4) | <0.001 |
| Living near waste disposal | 70 (37.2) | 12 (25.5) | 29 (55.8) | 29 (32.6) | 0.004 |
| Attending daycare | 4 (2.1) | 2 (4.3) | 1 (1.9) | 1 (1.1) | 0.374 |
| Medical history (%) | | | | | |
| Developmental delay | 52 (27.7) | 16 (34.0) | 21 (40.4) | 15 (16.8) | 0.003 |
| Congenital heart disease | 49 (26.1) | 16 (34.0) | 24 (46.2) | 9 (10.1) | <0.001 |
| Low birth weight | 46 (24.4) | 12 (25.5) | 20 (38.5) | 14 (15.7) | 0.011 |
| Severe malnutrition‡ | 35 (18.6) | 10 (21.3) | 13 (25.0) | 12 (13.5) | 0.205 |
| Premature baby | 34 (18.1) | 4 (8.5) | 16 (30.8) | 14 (15.7) | 0.012 |
| Neurological disorder | 25 (13.3) | 5 (10.6) | 17 (32.7) | 3 (3.4) | <0.001 |
| Tuberculosis (recent/cured) | 10 (5.3) | 4 (8.5) | 2 (3.8) | 4 (4.5) | 0.588 |
| Asthma | 9 (4.8) | 3 (6.4) | 1 (1.9) | 5 (5.6) | 0.563 |
| HIV disease§ | 2 (1.1) | 1 (2.1) | 1 (1.9) | 0 (0) | 0.315 |
| Immunisation history, fully vaccinated for age¶ (%): | | | | | |
| Measles | 141 (75.0) | 38 (80.9) | 41 (78.8) | 62 (69.7) | 0.175 |
| DPT-Hib | 105 (55.9) | 30 (63.8) | 25 (48.1) | 50 (56.2) | 0.233 |
| Pneumococcus | 4 (2.1) | 0 (0) | 4 (7.7) | 0 (0) | 0.009 |
| Influenza | 2 (1.1) | 0 (0) | 2 (3.8) | 0 (0) | 0.132 |
| Symptoms and signs (%) | | | | | |
| Shortness of breath | 174 (92.6) | 41 (87.2) | 48 (92.3) | 85 (95.5) | 0.214 |
| Cough | 171 (91.0) | 40 (85.1) | 42 (80.8) | 89 (100) | <0.001 |
| Intercostal retraction | 171 (91.0) | 43 (91.5) | 52 (100) | 76 (85.4) | 0.005 |
| Rhonchi | 168 (89.4) | 42 (89.4) | 39 (75.0) | 87 (97.8) | <0.001 |
| Fever | 152 (80.9) | 34 (72.3) | 35 (67.3) | 83 (93.3) | <0.001 |
| Chest indrawing | 125 (66.5) | 36 (76.6) | 43 (82.7) | 46 (51.7) | <0.001 |
| Fast breathing | 80 (42.6) | 15 (31.9) | 43 (82.7) | 22 (24.7) | <0.001 |

Continued

**Table 1** Continued

| Demographic characteristics | All (N=188) | Semarang (N=47) | Yogyakarta (N=52) | Tangerang (N=89) | P value |
|---|---|---|---|---|---|
| SpO$_2$ <90% and/or cyanosis | 43 (22.9) | 7 (14.9) | 17 (32.7) | 19 (21.3) | 0.098 |
| Diarrhoea | 36 (19.1) | 6 (12.8) | 4 (7.7) | 26 (29.2) | 0.003 |
| Wheezing | 35 (18.6) | 9 (19.1) | 10 (19.2) | 16 (18.0) | 1 |
| Vomiting | 14 (7.4) | 4 (8.5) | 5 (9.6) | 5 (5.6) | 0.595 |
| Inability to drink | 13 (6.9) | 4 (8.5) | 5 (9.6) | 4 (4.5) | 0.425 |
| Decreased consciousness | 7 (3.7) | 1 (2.1) | 1 (1.9) | 5 (5.6) | 0.612 |
| Seizure | 6 (3.2) | 1 (2.1) | 0 (0) | 5 (5.6) | 0.203 |
| Leucocyte count, median (IQR)×10$^3$/uL | 14.0 (10.4–18.9) | 14.9 (11.1–18.8) | 12.1 (9.8–17.8) | 14.0 (10.4–19.0) | 0.356 |
| Neutrophil-lymphocyte ratio, median (IQR) | 1.4 (0.9–2.8) | 1.3 (0.9–2.6) | 1.0 (0.6–2.0) | 1.9 (1.1–3.2) | 0.367 |
| CRP, median (IQR) mg/L | 9.0 (3.6–28.0) | 11.8 (1.6–23.3) | 9.0 (4.9–21.8) | 8.4 (1.5–34.1) | 0.665 |
| PCT, median (IQR) ng/mL | 0.2 (0.1–1.7) | 0.2 (0.1–1.5) | 0.2 (0.1–1.0) | 0.2 (0.1–2.6) | 0.912 |
| Severe pneumonia (WHO Classification 2014 version) (%) | 89 (47.3) | 26 (55.3) | 26 (50.0) | 37 (41.6) | 0.281 |
| CXR findings (%): | | | | | |
| Interstitial infiltrate | 131 (69.7) | 26 (55.3) | 30 (57.7) | 75 (84.3) | <0.001 |
| Alveolar infiltrate | 125 (66.5) | 41 (87.2) | 44 (84.6) | 40 (44.9) | <0.001 |
| Pleural effusion | 5 (2.7) | 1 (2.1) | 2 (3.8) | 2 (2.2) | 0.85 |
| Antibiotic administration prior to blood culture (%) | 150 (79.8) | 39 (83.0) | 49 (94.2) | 62 (69.7) | 0.002 |

*Low education of parents was defined by highest level of parents' formal education being high school diploma or less.
†A densely populated neighbourhood was defined as >200 people/km$^2$ or <8 m$^2$/person in the subject's home.
‡Severe malnutrition was defined as weight for height below −3 SD from the median of the WHO Child Growth Standards.
§Subjects were tested for HIV infection if a parent/guardian provided consent and a specimen was available (n=160).
¶Full vaccination was defined as being up to date for age per vaccination schedule at study enrolment.
CRP, C reactive protein; CXR, chest X-ray; PCT, procalcitonin.

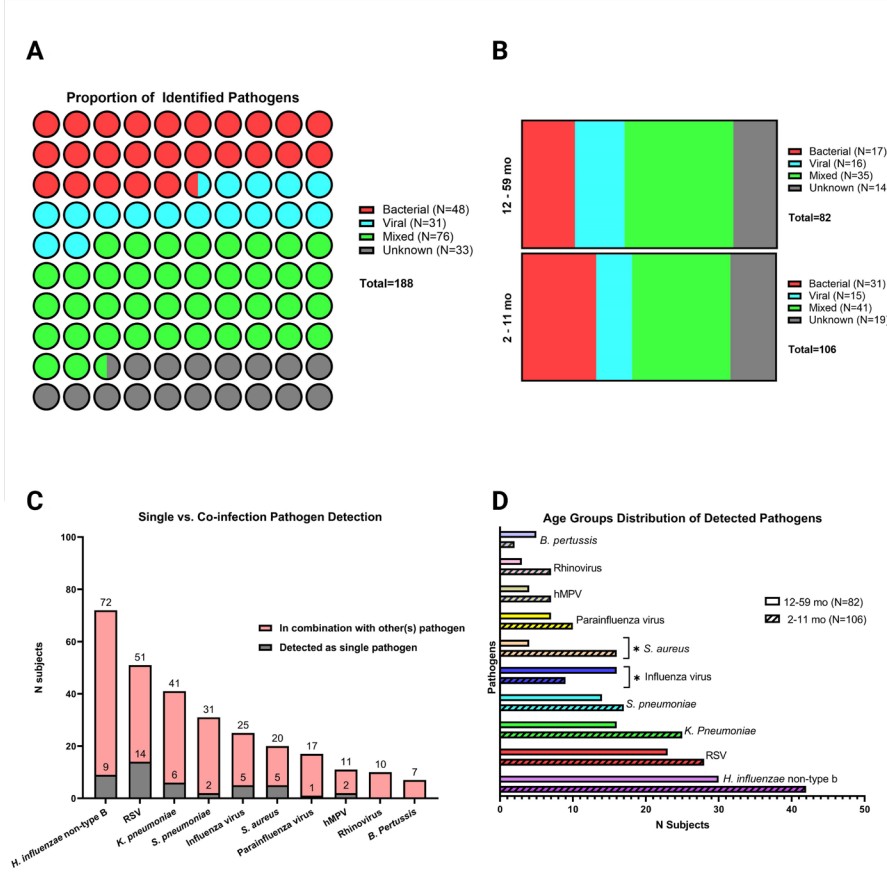

**Figure 2** Pathogen distribution. (A) Overall proportion of identified viral/bacterial/mixed pathogen, (B) Viral/bacterial/mixed pathogens by age group, (C) pattern of detection of the ten most identified pathogens, (D) distribution of 10 most identified pathogens by age group. *P<0.05. hMPV, human metapneumovirus; RSV, respiratory syncytial virus.

received 1–2 doses of antibiotics prior to collection of blood culture in the emergency unit, with the combination of ampicillin and gentamicin (34.6%), cefotaxime (17.0%) and ceftriaxone (14.4%) being the three most frequent regimens used. Details of antibiotic regimens administered before blood culture, including dosage and given frequency, are presented in online supplemental table 2.

### Detection of pathogens

Blood and sputum cultures were performed on specimens from 184 (97.9%) and 183 (97.3%) subjects, respectively. A total of 150 (79.8%) children received antibiotics prior to collection of blood for culture. Seventy-five (41.0%) sputum culture isolates were analysed from specimens meeting the required quality criteria. An NP or OP swab was obtained from 187 (99.5%) subjects, IS for PCR from 176 (93.6%), whole blood for PCR from 163 (86.7%) and paired acute-convalescent serum specimens for serology from 116 (61.7%) (figure 1).

The PEER-PePPeS algorithm was used to determine the causative pathogen(s) from those identified by culture, molecular and serological assay. Among the 188 study participants, 48 (25.5%) had bacterial infection, 31 (16.5%) had viral infection, 76 (40.4%) were of mixed bacterial and viral aetiology, and 33 (17.6%)

were of unknown aetiology (figure 2A). Mixed infection, the most common overall aetiology, was seen in 38.7% of 2–11 months and in 42.7% of 12–59 months (figure 2B). Mixed infection was also the predominant aetiology across all study sites (online supplementary file 2). *H. influenzae* non-type B (N=73, 38.8%), RSV (N=51, 27.1%), *K. pneumoniae* (N=43, 22.9%), *S. pneumoniae* (N=29, 15.4%), influenza virus (N=25, 13.3%), *S. aureus* (N=20, 10.6%), PIV (N=17, 9.0%), hMPV (N=11, 5.8%), Rhinovirus (N=10, 5.3%) and *B. pertussis* (N=7, 3.7%) were the top ten pathogens identified, more commonly appearing in mixed infection as opposed to as a sole pathogen (figure 2C). Influenza virus was significantly higher in the age group 12–59 months vs 2–11 months (N=16, 64%, p=0.027), while *S. aureus* was significantly more common in 2–11 months vs 12–59 months (N=16, 80%, p=0.024). Though not statistically significant, other pathogens trended toward more frequent detection in age group 2–11 mo (except *B. pertussis*) (figure 2D). Among 76 mixed infection cases, RSV +*H. influenzae* non-type B was the most common coinfection (N=22, 28.9%), followed by RSV +*S. pneumoniae* (N=10, 13.2%), influenza virus +*H. influenzae* non-type B (N=10, 13.2%), RSV +*K. pneumoniae* (N=9, 11.8%) and PIV+*H. influenzae* non-type B (N=9, 11.8%) (data not shown).

We observed no difference in pathogen distribution by pneumonia severity based on WHO classification system (online supplemental table 3 and online supplemental fig 3). By pathogen, there was no significant difference in distribution between pneumonia severity status or mortality, except for *S. pneumoniae* which was found in significantly more severe cases using the WHO system (p=0.033) (online supplemental table 3).

A comparison of positivity rates for each causative pathogen by detection method is shown in table 2. Overall, PCR captured more bacterial pathogens than culture and more viral pathogens than acute-convalescent paired serology. Paired serology was generally helpful in identifying atypical bacteria, such as *C. pneumoniae* and *L. pneumophila* and upper respiratory tract viruses, such as Rhinovirus and Enterovirus. When comparing blood and IS culture, IS yielded more positive bacterial pathogen results. Similarly, IS PCR captured more pathogens than NP/OP PCR.

## Mortality

Nineteen (10.1%) of the 188 subjects died during the 30-day study period. Seven (36.8%) of these 19 were male, and most (N=17, 89.5%) were less than 1 year old. Among the 19 deceased subjects, median study duration was 12 (IQR 4–17.5) days; 8 (42.1%) were admitted to ICU and 6 (31.6%) received mechanical ventilation. Twelve (63.2%) died due to respiratory failure, three (15.8%) due to sepsis and three (15.8%) for unknown reasons after discharge (data not shown). Most deaths occurred in the 2–11 mo age group compared with the 12–59 mo age group (78.9% vs 21.1%, p=0.036). Causative pathogens for deceased subjects were bacterial-only in seven (36.8%), viral-only in two (10.5%), mixed in five (26.3%) and unknown in five subjects (26.3%). There were no significant differences in pathogen distribution between subjects that survived and died. *H. influenzae* non-type B was the most common pathogen identified in deceased subjects (N=8, with the case fatality rate (CFR) in this study of 11.0%), followed by *K. pneumoniae* (N=6, CFR of 13.9%), influenza virus (N=3, CFR of 12.0%), *B. pertussis* (N=2, CFR of 28.6%) and RSV (N=2, CFR of 3.9%) (online supplemental table 3). Pre-existing conditions among deceased subjects included congenital heart disease (N=10, 52.6%), severe malnutrition (N=7, 36.8%) and developmental delay (N=7, 36.8%). A clinical summary of the fatal cases is shown in online supplemental table 4.

## Seasonality

During the 27-month study period, infections caused by RSV and influenza were seen year-round with peak activity occurring during the wet season (November to March) in Indonesia (66.7%, p<0.001; and 64.0%, p=0.012, respectively). However, there was little variation in detection of the most common respiratory bacterial infections by month and season. *H. influenzae* non-type B shows peaks in August (N=12, 16.4%) and March (N=11, 15.1%), while *K. pneumoniae* and *S. pneumoniae* fluctuate at lower levels throughout the year (figure 3).

## DISCUSSION

PEER-PePPeS, a prospective multisite study, characterised the current epidemiology of CAP in children 2–59 months in Indonesia. No recent prospective Indonesian studies address this topic. Our study found: (1) mixed bacterial and viral infection is the most frequent (N=76, 40.4%) cause of childhood CAP, irrespective of age group and pneumonia severity; (2) bacterial infections were common (66% of cases) with *H. influenzae* non-b type, *K. pneumoniae* and *S. pneumoniae* as the three most common bacterial aetiologies; (3) viral pathogens were also common (57% of PEER-PePPeS subjects), with 16.5% of cases attributed to virus only and RSV and Influenza Virus being the most common viruses identified and (4) PCR on IS specimens was the most sensitive assay for pathogen identification.

While our findings are consistent with other studies, clinical significance of mixed infection remains controversial. It is unclear if both agents act as true pathogens.[22 26] PEER-PePPeS did not demonstrate a correlation of mixed infection with pneumonia severity and 30-day mortality. Many deaths occurred at a younger age (<1 year old) and with comorbidities, such as congenital heart disease and severe malnutrition, similar to previous reports.[27 28] Such factors should be considered in prevention and management of childhood pneumonia to reduce mortality rate.

In recent years, there has been an increased focus on the role of respiratory viruses in childhood pneumonia, partly attributable to use of conjugate pneumococcal and Hib vaccines and increased detection by PCR.[21 22 29 30] In PEER-PePPeS, viruses were found in 57% of subjects (virus only +mixed infection), with 16.5% of cases attributed to virus only. Thus, many patients probably received unnecessary antibiotics when covered empirically per current Indonesian guidelines. Improving ability to discriminate between viral and bacterial infections would facilitate optimisation of antibiotic administration and counter antimicrobial resistance.[31]

RSV and influenza virus were the most commonly detected viruses in this study and may be associated with Indonesia's wet/rainy season.[32–34] A high prevalence of RSV was also observed in the GABRIEL and PERCH international case–control studies of childhood pneumonia aetiology.[22 30] In terms of mixed infections, we found that RSV +*H. influenzae* non-type B and RSV +*S. pneumoniae* were most common. Since respiratory viruses such as RSV can predispose to secondary bacterial infections, particularly *S. pneumoniae* and *H. influenzae*,[35] and conversely bacteria can increase RSV susceptibility,[35 36] these coinfections highlight the need for optimising RSV surveillance, prevention and treatment.

Though influenza virus also increases risk for secondary bacterial infections and is a major cause of childhood morbidity and mortality worldwide, data from developing

**Table 2** Causative pathogens per PEER-PePPeS rules by detection method

| Pathogen | N | Blood culture N (%) | IS culture N (%) | Whole blood PCR N (%) | NP / OP PCR N (%) | IS PCR N (%) | Serology test N (%) |
|---|---|---|---|---|---|---|---|
| Gram-positive cocci bacteria | | | | | | | |
| Streptococcus pneumoniae | 29 | 1 (3.4) | 3 (10.3) | -- | 21 (72.4) | 28 (96.6) | |
| Staphylococcus aureus | 20 | -- | 7 (35) | -- | 11 (55) | 19 (95) | |
| Streptococcus mitis | 4 | -- | 4 (100) | | | | |
| Streptococcus pyogenes | 1 | -- | 1 (100) | | | | |
| Gram-negative cocci bacteria | | | | | | | |
| Moraxella catarrhalis | 2 | -- | 2 (100) | | 2 (100) | 2 (100) | |
| Gram-negative rods bacteria | | | | | | | |
| Haemophilus influenzae non-type b | 73 | -- | -- | 8 (10.9) | 60 (82.2) | 71 (98.6) | |
| Klebsiella pneumoniae | 43 | -- | 17 (39.5) | | 2 (4.7) | 34 (79.1) | |
| Bordetella pertussis | 7 | -- | -- | | | 7 (100) | |
| Escherichia coli | 5 | 1 (20) | 4 (80) | | | | |
| Pseudomonas aeruginosa | 4 | -- | 4 (100) | | | | |
| Acinetobacter baumannii | 3 | -- | 3 (100) | | | | |
| H. inf type b | 2 | -- | -- | -- | -- | 2 (100) | |
| Neisseria meningitidis | 1 | 1 (100) | 1 (100) | | | | |
| Atypical-bacteria | | | | | | | |
| Chlamydia pneumoniae | 5 | -- | -- | | | -- | 5 (100) |
| Mycoplasma pneumoniae | 5 | -- | -- | | | 5 (100) | 1 (20) |
| Legionella pneumophila | 1 | -- | -- | | | -- | 1 (100) |
| Virus | | | | | | | |
| RSV | 51 | | | | 36 (70.6) | 45 (88.2) | 10 (19.6) |
| RSV A | 15 | | | | 10 (66.7) | 13 (86.7) | |
| RSV B | 36 | | | | 26 (72.2) | 32 (88.8) | |
| Influenza virus | 25 | | | | 16 (64) | 22 (88) | 9 (36) |
| inf A (H1N1) | 7 | | | | 7 (100) | 7 (100) | 7 (70) |
| inf A (H3N2) | 3 | | | | 3 (100) | 3 (100) | |
| inf B | 14 | | | | 6 (42.9) | 12 (85.7) | 2 (14.3) |
| PIV | 17 | | | | 16 (94.1) | 15 (88.2) | 3 (17.6) |

Continued

**Table 2** Continued

| Pathogen | N | Blood culture N (%) | IS culture N (%) | Whole blood PCR N (%) | NP / OP PCR N (%) | IS PCR N (%) | Serology test N (%) |
|---|---|---|---|---|---|---|---|
| PIV 1 | 5 | | | | 5 (100) | 4 (80) | 3 (17.6) |
| PIV 2 | 0 | | | | -- | -- | |
| PIV 3 | 11 | | | | 10 (90.9) | 10 (90.9) | |
| PIV 4 | 1 | | | | 1 (100) | 1 (100) | |
| hMPV | 11 | | | | 5 (45.5) | 10 (90.9) | |
| Rhinovirus | 10 | | | | 10 (100) | 6 (60) | 4 (40) |
| Enterovirus | 5 | | | | 3 (60) | 3 (60) | 3 (60) |
| Bocavirus | 3 | | | | 2 (66.7) | 3 (100) | |
| hCoV-NL63 | 2 | | | | 2 (100) | 2 (100) | |

Grey box indicates the assay was not performed.

hMPV, human metapneumovirus; IS, induced sputum; NP, nasopharyngeal; PIV, parainfluenza virus; RSV, respiratory syncytial virus.

countries is scarce.[37] In a previous Indonesian study of hospitalised patients with a SARI, the prevalence of the influenza virus was 10.6% in children under 5 years old, and was never diagnosed during hospitalisation.[38] PEER-PePPeS confirms the need for improved diagnostic strategies, management optimisation and influenza vaccination in children. Of note, our study was conducted before identification of COVID-19 in Indonesia,[39] so did not address the role of COVID-19 in childhood pneumonia.

We also found that 66% of cases were caused by bacterial infection (bacteria only +mixed infection). Overall, *H. influenzae* non-type B was the most common bacteria implicated, followed by *K. pneumoniae* and *S. pneumoniae*. *H. influenzae* non-type B predominance was also observed in a Malaysian study, where 90% of enrolled children were vaccinated against Hib as part of the national immunisation programme.[24] With Indonesia's moderate (56.4%) Hib vaccine coverage, high incidence of *H. influenzae* non-type B may represent its true prevalence or strains not covered by Hib vaccine.[40] This finding agrees with current data that non-typeable *H. influenzae* (NTHi) can cause significant illness, and argues for strengthening paediatric diagnostic laboratory capacity.

Our identification of *K. pneumoniae* as the second most common bacterial aetiology is consistent with high carriage rates (~7%) in healthy Indonesian children. Carriage has been related to poor food and water sanitation and may give rise to pneumonia, especially in children with malnutrition.[41] Given *K. pneumoniae's* potential for antibiotic resistance and high virulence of some strains, proactive detection and management strategies should be prioritised.[42]

The relatively low prevalence (15.4%) of *S. pneumoniae* in PEER-PePPes was surprising since carriage rates are high and PCV coverage low in Indonesia.[43] Low prevalence has also been reported from Malaysia, where PCV coverage is 8.7%[24] and in the PERCH study, reflecting temporal shifts in childhood pneumonia aetiologies.[22] As only 4.8% of PEER-PePPeS subjects had received PCV, vaccination alone cannot account for the low *S. pneumoniae* prevalence. Antibiotic exposure prior to specimen collection may have reduced colonisation density and lowered the yield of *S. pneumoniae* by both culture and PCR.[44] Moreover, our panel did not include *S. pneumoniae* paired serology, which may be useful to increase pneumococcal diagnosis in young children.[45] Nonetheless, *S. pneumoniae* remains an important aetiological agent of severe/complicated CAP globally.[46] Our finding that *S. pneumoniae* was significantly associated with severe cases by the WHO classification system supports the need for ongoing surveillance, vaccination and prevention of transmission between adults and children.

Inclusion of several pathogen identification strategies in PEER-PePPes demonstrates the differential utility of assays and specimen types. Our findings highlight the value of molecular assays, especially in culture-negative cases where microorganisms may be nonrecoverable in culture due to prior antibiotics or presence of otherwise

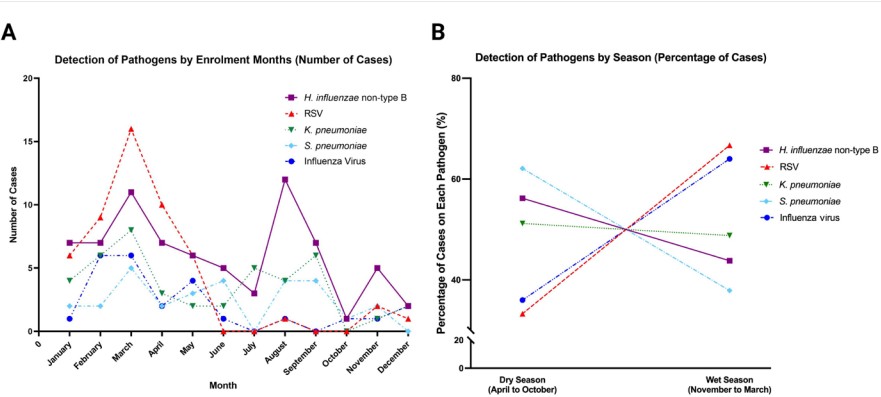

**Figure 3** Distribution of the (A) monthly count and (B) seasonal pattern of infection caused by *Haemophilus influenzae* non-type B, RSV, *Klebsiella pneumoniae*, *Streptococcus pneumoniae* and influenza virus during a 27-month study period. RSV, respiratory syncytial virus.

difficult to culture bacteria.[47 48] PCR is also less laborious and can identify genes associated with antibiotic resistance, though conventional culture methods are required to confirm phenotypic resistance.[49 50] Even with the limited PCR panels used in our study, molecular assays had greater sensitivity for identification of bacterial pathogens than blood or sputum culture.

Although sensitive for detection, PCR does not provide information regarding infectiousness or viability. Genome fragments from dead organisms may be detected, often at a low level, even after clinical resolution.[48] Furthermore, negative results may occur due to differential viral kinetics along the respiratory tract. Lower respiratory tract specimens, such as IS, should be sought as they originate from the site of infection.[12 13] Accordingly, we observed a higher yield from PCR on IS than NP specimens. We also found that the use of paired serologies increased the diagnostic yield and was useful for pathogen confirmation, particularly in the setting of innocent bystander viruses and atypical bacteria.[12]

PEER-PePPeS used a comprehensive approach for pathogen detection to increase diagnostic yield. It also enrolled patients over a 27-month study period, facilitating assessment of seasonality. However, our study has several limitations. First, the relatively small sample size, and observational design may limit generalisability and causal inference. Second, most subjects (79.8%) received antibiotics before specimen collection in accordance with national guidelines. To address this, we enrolled subjects within 24 hours of admission, and specimens were collected as soon as possible to minimise the effects of antibiotics on culture results. Third, we did not enrol healthy control children, limiting the ability to estimate the adjusted population attributable fraction of each pathogen.[29 30] A healthy control group could have revealed baseline carriage rates, minimising overattribution of disease to non-pathogenic organisms.[21 22 29 30] Fourth, we did not collect lung aspirates or pleural fluid specimens, which are superior for determination of pneumonia aetiology.[15] Fifth, several subjects had pneumonia

of unknown aetiology; this may have been due to administration of antibiotics before culture which could reduce sensitivity, poor IS quality, the limited panel of bacterial and viral pathogens tested, lack of fungal testing, or currently unrecognised causes of paediatric pneumonia.

In conclusion, the epidemiology of childhood CAP is constantly evolving in step with social and environmental factors and thus, should be regularly assessed. Our study found that *H. influenzae* non-type B and RSV were the most common pathogens causing hospitalised CAP among Indonesian children aged 2–59 months old, reflecting temporally dynamic aetiologies of childhood CAP; studies from the 1970s–1990s mainly detected *S. pneumoniae* and *H. influenzae* type B as the most important causes of childhood pneumonia in LMICs.[3–5] PCR on IS demonstrated the best sensitivity for pathogen identification. We recommend incorporating molecular assays for pathogen detection, preferably multiplexed point-of-care assays, into practice guidelines. Improvements in Indonesia's lab infrastructure during the COVID-19 pandemic can be leveraged to facilitate use of molecular assays for evaluation of childhood CAP. Optimisation of pathogen detection to understand changing childhood CAP epidemiology will also inform public policy on prevention and management.

**Author affiliations**
[1]Tangerang District General Hospital, Tangerang, Banten, Indonesia
[2]Rumah Sakit Umum Pusat Dr Kariadi, Semarang, Central Java, Indonesia
[3]Rumah Sakit Umum Pusat Dr Sardjito, Sleman, DIY, Indonesia
[4]Indonesia Research Partnership on Infectious Disease, Jakarta, Indonesia
[5]National Institute of Health Research and Development, Ministry of Health, Republic of Indonesia, Jakarta, Indonesia
[6]National Institute of Allergy and Infectious Diseases, Bethesda, Maryland, USA
[7]National Cancer Institute, Bethesda, Maryland, USA

**Contributors** DL, HF, RT, YM, HK, AMN, ATA, C-YL and CL designed and conceptualised the study. DL, HF, RT, AB, CH, MSA, DW, MM, SD and AS performed clinical assessments and were responsible for data entry. DL, HF, RT, YM, HK, AMN, NL, AK and C-YL designed the methodology for pathogen identification. YM, HK and AMN performed data analysis, interpretation and drafted the first manuscript.

DL, HK, ATA, MK, AN, C-YL and CL assisted with manuscript writing, analysis and interpretation of data. All authors contributed to manuscript development, edited for critical conten and have approved the final version. HK acts as guarantor for the final manuscript.

**Funding** This manuscript has been funded in whole or in part with MoH Indonesia, National Academy of Sciences (Sub-Grant Number: 2000007599), and Federal funds from the NIAID, NIH, under contract Nos. HHSN261200800001E and HHSN261201500003I.

**Disclaimer** The content of this publication does not necessarily reflect the views or policies of the Department of Health and Human Services, nor does mention of trade names, commercial products, or organisations imply endorsement by the U.S. Government.

**Competing interests** None declared.

**Patient and public involvement** Patients and/or the public were not involved in the design, or conduct, or reporting, or dissemination plans of this research.

**Patient consent for publication** Not applicable.

**Ethics approval** This study was approved by the Ethical Clearance Committee of Faculty of Medicine, Universitas Indonesia (No. 567/UN2.F1/ETIK/2017). The study was conducted in accordance with the Declaration of Helsinki. Written informed consent was obtained from parents or guardians before enroment. Participants gave informed consent to participate in the study before taking part.

**Provenance and peer review** Not commissioned; externally peer reviewed.

**Data availability statement** Data are available on reasonable request. Data are available on reasonable request. The anonymised data set will be shared following the signing of a data-sharing agreement, with permission of the ethical clearance committee, study authors and all project partners, exclusively for non-commercial purposes. Please contact the corresponding author with any queries.

**ORCID iD**
Yan Mardian http://orcid.org/0000-0002-4113-5273

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
