## [Reviewer comments · BMJ Open]

ARTICLE DETAILS

TITLE (PROVISIONAL)	Epidemiology of community-acquired pneumonia among hospitalized children in Indonesia: a multicenter, prospective study
AUTHORS	Lokida, Dewi; Farida, Helmia; Triasih, Rina; Mardian, Yan; Kosasih, Herman; Naysilla, Adhella Menur; Budiman, Arif; Hayuningsih, Chakrawati; Anam, Moh. Syarofil; Wastoro, Dwi; Mujahidah, Mujahidah; Dipayana, Setya; Setyati, Amalia; Haman, Abu; Lukman, Nurhayati; Karyana, Muhammad; Kline, Ahnika; Neal, Aaron; Lau, Chuen-Yen; Lane, C.

VERSION 1 – REVIEW

REVIEWER	N. Haddad, Raymond Necker-Children's Institute, Pediatric and Congenital Cardiology
REVIEW RETURNED	23-Nov-2021

GENERAL COMMENTS	Abstract: 1. Please add the % after each absolute value2. Define IS3. The conclusion is not quite accurate based on the results section and needs to be changed. 40.4% is not quite different from 25.5% and 16.5%, especially since 17.8% of cases were not classified. The strengths and limitations of the study are well described yet the limitations are real and I would expect to overcome this with a prospective study to present more solid conclusions than previously published reports. Introduction: the 2 first paragraphs are too long and need to be shortened to half. Methods: 1. Lines 114 to 118 are not really needed and can be outlined in a drawing.2. Lines 118 to 120 are to remove3. Lines 143-144: was "not" practical4. Lines 160: define NP and IS Stats: I suppose that continuous data is not normally distributed. Why t-test was used instead of Mann-Whitney U (or Kruskal-Wallis) tests P-values are interesting to add in table 1 to prove the non-statistical significance of homogeneity of study data across the 3 recruiting centers. Lines 230-238 should be also outlined in Table 1 for a clearer reading
---

	Lines 240: A huge number of patients received antibiotics before the culture >> This is a major limitation to discuss. Why did those patients receive antibiotics although it was a prospective study?? Lines 279: Mortality numbers are quite high. Please describe more what happened and more specifically the demographics of dead patients. Lines 377-378 are the most important sentences of this study and I would have expected to solve this issue in a prospective study. If cultures/PCR results do not impact care, what is specifically its clinical and practical usefulness?? Lines 397-402: how can you be sure that detected pathogens are causative pathogens?? A huge number of patients received antibiotics before culture/swab >> this needs to be seriously discussed Impact on clinical care and guidelines derived from study findings should be outlined in the conclusions. this is the message we are waiting for. Add this paper to the references: Tannous R, Haddad RN, Torbey PH. Management of Community-Acquired Pneumonia in Pediatrics: Adherence to Clinical Guidelines. Front Pediatr. 2020;8:302. Published 2020 Jun 19. doi:10.3389/fped.2020.00302
--	--

REVIEWER	Chang, Tu-Hsuan Chi Mei Medical Center, Department of Pediatrics
REVIEW RETURNED	12-Feb-2022

GENERAL COMMENTS	Thank you for inviting me to review this manuscript. This is a prospective study using comprehensive approach to investigate common CAP pathogens in Indonesia children. The article is well written and the method is rigorous. It provided valuable epidemiological data. I really appreciate the work done by all investigators. I have just a few comments and suggestions.  1. To support the description of "subject characteristics were similar across the three study sites", related statistical analysis should be presented in table 1. 2. The first letter of virus should be a lower case. 3. Some abbreviations (e.g.: "IS" specimens in abstract) need to be spell out when first appeared.
---

VERSION 1 – AUTHOR RESPONSE

Reviewer: 1

Dr. Raymond N. Haddad, Necker-Children's Institute

Competing interests of Reviewer: I have no conflict of interest to declare

We would like to thank the reviewer for the constructive and helpful comments. We have carefully considered the reviewer's comments and provide an itemized response below

Comments to the Author:

Abstract:

1. Please add the % after each absolute value

Thank you for your suggestion. **Lines 49-51** now read:

“The five most common causative pathogens identified were *Haemophilus influenzae* non-type B (N=73, **38.8%**), respiratory syncytial virus (RSV) (N=51, **27.1%**), *Klebsiella pneumoniae* (N=43, **22.9%**), *Streptococcus pneumoniae* (N=29, **15.4%**), and Influenza virus (N=25, **13.3%**).”

2. Define IS

Thank you for your suggestion. **Lines 53-54** now read:

“The polymerase chain reaction (PCR) assays on **induced sputum** (IS) specimens captured most of the pathogens identified in this study.”

3. The conclusion is not quite accurate based on the results section and needs to be changed. 40.4% is not quite different from 25.5% and 16.5%, especially since 17.8% of cases were not classified.

Thank you for your correction. **Lines 56-58** now read:

“Our study found that *H. influenzae* non-type B and RSV **were the most frequently identified** pathogens causing hospitalized CAP among Indonesian children aged 2-59 months old. Our study also highlights the importance of PCR for diagnosis and by extension, appropriate use of antimicrobials.”

The strengths and limitations of the study are well described yet the limitations are real and I would expect to overcome this with a prospective study to present more solid conclusions than previously published reports.

Thank you for your comments. We agree that our findings are strengthened by the prospective design of our study. We have now emphasized this point in the discussion (**lines 310-311**). We also agree that the limitations are real. Our study (PEER-PePPes) was a prospective cohort study, which is an observational study and does not include intervention to the subjects. As such, we could not control the medical management of subjects, including the administration of antibiotics, since it has become the standard of care in hospitals based on Indonesia's current practical guideline. In addition, the molecular and serology assays were not conducted as a standard of care in hospital sites. Instead, we performed them retrospectively in INA-RESPOND Reference Lab (described in **Line-173-175**), so the treatment given during hospitalization was not based on the findings of this study. In current version, we add one more item in the "strengths and limitations of this study" section: *The relatively small sample size, geographic limitation to the island of Java and observational design may limit generalizability and causal inference*. A further explanation for antibiotics administration prior to the specimens' collection in this study will be provided in the below comments.

Introduction: the 2 first paragraphs are too long and need to be shortened to half.

Thank you for your suggestion. **Lines 79-89** now read:

“Pneumonia is the leading infectious cause of child mortality, with a greater burden in low- and middle-income countries (LMICs).[1] In Indonesia, pneumonia contributed to 15% of childhood deaths and was the second leading cause of death amongst children under five years in 2017.[2] Indonesian practice guidelines are adapted from the World Health Organization (WHO) guidelines, which are based on 1970's – 1990's data showing bacteria such as *Haemophilus influenzae* type b (Hib) and *Streptococcus pneumoniae* caused the majority of fatal pneumonias in children.[3–5] Therefore empiric antibiotics are considered first-line treatment for children with community-acquired pneumonia (CAP).[6–8] Despite

evidence that appropriate antibiotics are lifesaving, rational selection of antibiotics for pneumonia is hampered by low adherence to guidelines and scarcity of point-of-care diagnostics.[9–11] Consequently, healthcare providers, particularly those in LMIC, are likely to overtreat non-bacterial pneumonia with antibiotics.[11,12]”

Methods:

1. Lines 114 to 118 are not really needed and can be outlined in a drawing.
Thank you. The details of satellites sites have been removed from the text. They are depicted in Supplementary Figure 1.
2. Lines 118 to 120 are to remove
Thank you. We have removed that section.
3. Lines 143-144: was "not" practical
Thank you for the correction. **Lines 138-142** now read:
Children were excluded if they had been hospitalized for >24 hours; had a malignancy or history of malignancy; a history of long term (>2 months) steroid therapy; or conditions that might interfere with compliance with the study procedures (e.g., very ill patients for whom specimens could not be obtained or living outside the area for which follow-up was **not** practical).
4. Lines 160: define NP and IS
Thank you for the comment. These abbreviations (nasopharyngeal and induced sputum) are defined in the preceding paragraph (**line 149**).

Stats: I suppose that continuous data is not normally distributed. Why t-test was used instead of Mann-Whitney U (or Kruskal -Wallis) tests.

Thank you for your correction. We have modified the statistical methods. **Lines 195-198** now read:

Differences in categorical variables were compared using Pearson χ^2 or Fisher's exact test when the expected values in any of the contingency table cells are below 5. Differences in continuous variables were compared using One-way ANOVA or Kruskal-Wallis H-test for the data which does not follow the normal distribution based on Levene's test.

P-values are interesting to add in table 1 to prove the non-statistical significance of homogeneity of study data across the 3 recruiting centers.

Thank you for your input. We modified table 1 as follows, and also revised some sentences in the text based on statistically significant results.

Lines 216-223 now read:

Demographic and clinical characteristics are presented in **Table 1**. Overall, **several parameters, such as age, gender, laboratory, and WHO classification of pneumonia severity, were similar across the three study sites**. The median age was nine months (IQR, 5 to 20), and 54.7% of subjects were male. The most common comorbid conditions were developmental delay (27.7%), congenital heart disease (26.1%), and severe malnutrition (18.6%), with the subjects from Yogyakarta site having the greatest proportion of those comorbidities. The percentage of subjects who had been vaccinated (age-adjusted) against pneumococcus, influenza, Hib-DPT, and measles vaccines were 2.1%, 1.1%, 55.9%, and 75.0%, respectively.

Table 1 now reads:

Table 1. Baseline Characteristics of Subjects.

Demographic Characteristics	All (N=188)	Semarang (N=47)	Yogyakarta (N=52)	Tangerang (N=89)	P-value
Age, median (IQR) months	9 (5 – 20)	9 (5.5 – 21)	8 (4 – 13.3)	11 (5-20)	0.442
Gender, Male, (%)	103 (54.7)	29 (61.7)	26 (50)	48 (53.9)	0.493
Household Characteristics, (%):	163 (86.7)	37 (78.7)	43 (82.7)	84 (94.3)	0.019
• Low Education of Parents*	121 (64.4)	19 (40.4)	42 (80.8)	60 (67.4)	<0.001
• Living in a dense neighborhood†	70 (37.2)	12 (25.5)	29 (55.8)	29 (32.6)	0.004
• Living near waste disposal	109 (58.0)	22 (46.8)	43 (82.7)	44 (49.4)	<0.001
• Sick household contact <14 days	120 (63.8)	24 (51.1)	27 (51.9)	69 (77.5)	0.001
• Exposure to cigarette smoke	4 (2.1)	2 (4.3)	1 (1.9)	1 (1.1)	0.374
• Attending daycare					
Medical history (%):					
• Premature baby	34 (18.1)	4 (8.5)	16 (30.8)	14 (15.7)	0.012
• Low birth weight	46 (24.4)	12 (25.5)	20 (38.5)	14 (15.7)	0.011
• Developmental delay	52 (27.7)	16 (34.0)	21 (40.4)	15 (16.8)	0.003
• Congenital heart disease	49 (26.1)	16 (34.0)	24 (46.2)	9 (10.1)	<0.001
• Severe malnutrition‡	35 (18.6)	10 (21.3)	13 (25.0)	12 (13.5)	0.205
• Neurological disorder	25 (13.3)	5 (10.6)	17 (32.7)	3 (3.4)	<0.001
• Asthma	9 (4.8)	3 (6.4)	1 (1.9)	5 (5.6)	0.563
• HIV disease§	2 (1.1)	1 (2.1)	1 (1.9)	0 (0)	0.315
• Tuberculosis (recent/cured)	10 (5.3)	4 (8.5)	2 (3.8)	4 (4.5)	0.588
Immunization history, fully vaccinated for age (%):					
• DPT-Hib	105 (55.9)	30 (63.8)	25 (48.1)	50 (56.2)	0.233
• Influenza	2 (1.1)	0 (0)	2 (3.8)	0 (0)	0.132

Demographic Characteristics	All (N=188)	Semarang (N=47)	Yogyakarta (N=52)	Tangerang (N=89)	P-value
 • Pneumococcus • Measles 	4 (2.1) 141 (75.0)	0 (0) 38 (80.9)	4 (7.7) 41 (78.8)	0 (0) 62 (69.7)	0.009 0.175
Symptoms and signs (%):					
 • Cough • Shortness of breath • Fever • Decreased Consciousness • Inability to drink • Diarrhea • Vomiting • Seizure • Fast breathing • Intercostal retraction • Rhonchi • Wheezing • Chest indrawing • SpO₂ <90% and/or Cyanosis 	171 (91.0) 174 (92.6) 152 (80.9) 7 (3.7) 13 (6.9) 36 (19.1) 14 (7.4) 6 (3.2) 80 (42.6) 171 (91.0) 168 (89.4) 35 (18.6) 125 (66.5) 43 (22.9)	40 (85.1) 41 (87.2) 34 (72.3) 1 (2.1) 4 (8.5) 6 (12.8) 4 (8.5) 1 (2.1) 15 (31.9) 43 (91.5) 42 (89.4) 9 (19.1) 36 (76.6) 7 (14.9)	42 (80.8) 48 (92.3) 35 (67.3) 1 (1.9) 5 (9.6) 4 (7.7) 5 (9.6) 0 (0) 43 (82.7) 52 (100) 39 (75.0) 10 (19.2) 43 (82.7) 17 (32.7)	89 (100) 85 (95.5) 83 (93.3) 5 (5.6) 4 (4.5) 26 (29.2) 5 (5.6) 5 (5.6) 22 (24.7) 76 (85.4) 87 (97.8) 16 (18.0) 46 (51.7) 19 (21.3)	<0.001 0.214 <0.001 0.612 0.425 0.003 0.595 0.203 <0.001 0.005 <0.001 1.000 <0.001 0.098
Leukocyte count, median (IQR) × 10 ³ /uL	14.0 (10.4 – 18.9)	14.9 (11.1 – 18.8)	12.1 (9.8 – 17.8)	14.0 (10.4 – 19.0)	0.356
Neutrophil-lymphocyte ratio (NLR), median (IQR)	1.4 (0.9 – 2.8)	1.3 (0.9 – 2.6)	1.0 (0.6 – 2.0)	1.9 (1.1 – 3.2)	0.367
CRP, median (IQR) mg/L	9.0 (3.6 – 28.0)	11.8 (1.6 – 23.3)	9.0 (4.9 – 21.8)	8.4 (1.5 – 34.1)	0.665
PCT, median (IQR) ng/mL	0.2 (0.1 – 1.7)	0.2 (0.1 – 1.5)	0.2 (0.1 – 1.0)	0.2 (0.1 – 2.6)	0.912
Severe pneumonia (WHO Classification 2014 version) (%)	89 (47.3)	26 (55.3)	26 (50.0)	37 (41.6)	0.281
CXR Findings (%):					
 • Pleural effusion 	5 (2.7)	1 (2.1)	2 (3.8)	2 (2.2)	0.850

Demographic Characteristics	All (N=188)	Semarang (N=47)	Yogyakarta (N=52)	Tangerang (N=89)	P-value
• Interstitial infiltrate	131 (69.7)	26 (55.3)	30 (57.7)	75 (84.3)	<0.001
• Alveolar infiltrate	125 (66.5)	41 (87.2)	44 (84.6)	40 (44.9)	<0.001
Antibiotic administration prior to blood collection for blood culture (%):	150 (79.8)	39 (83.0)	49 (94.2)	62 (69.7)	0.002
• Ampicillin					
• Ampicillin – Gentamicin	14 (7.4)	5 (10.6)	9 (17.3)	0 (0)	
• Cefotaxime	64 (34.0)	25 (53.2)	39 (75.0)	0 (0)	
• Ceftriaxone	33 (17.5)	0 (0)	0 (0)	33 (37.1)	
• Others [¶]	26 (13.8)	0 (0)	0 (0)	26 (29.2)	
	13 (6.9)	9 (19.1)	1 (1.9)	3 (3.4)	

[†]Low education of parents was defined by highest level of parents' formal education being high school diploma or less; [‡]A densely populated neighborhood was defined as >200 people/km² or <8 m²/person in the subject's home; [‡] Severe malnutrition was defined as weight for height below -3 standard deviations from the median of the WHO Child Growth Standards; [§]Subjects were tested for HIV infection if a parent / guardian provided consent and a specimen was available (n=160); ^{||}Full vaccination was defined as being up to date for age per vaccination schedule at study enrollment; [¶]Other antibiotics include: Amoxicillin, Cefamandole, Ceftazidime, Gentamicin, Cefotaxime – Amikacin, Ceftriaxone – Gentamicin, and Azithromycin – Ampicillin – Gentamicin

Lines 230-238 should be also outlined in Table 1 for a clearer reading

Thank you, we now include signs and symptoms, laboratory parameters and antibiotic administration prior to blood collection for blood culture in Table 1 (described above)

Lines 240: A huge number of patients received antibiotics before the culture >> This is a major limitation to discuss. Why did those patients receive antibiotics although it was a prospective study??

Thank you, we appreciate your concern. This is a very important point. Our study (PEER-PePPes) was a prospective observational study. As such, we were not able to control the medical management of subjects, including the administration of antibiotics. Clinicians made these decisions taking into account available guidelines along with their experience and judgment. We agree that an ideal etiologic study should collect specimens before antibiotics or exclude subjects with previous antibiotic exposure. However, the Indonesian pediatric association practice guideline recommendation that antibiotics should be administered early in children hospitalized with pneumonia, prevented this ideal study approach. To overcome this, our inclusion criteria were to enroll subjects within 24 hours of admission, and specimens were collected as soon as possible to hopefully minimize the effects of antibiotics on culture results. This is just an inherent limitation of the observational nature of this study. We also now add this limitation to the discussion section.

Lines 399-405 now read:

“The relatively small sample size, geographic limitation to the island of Java and observational design may limit generalizability and causal inference. Second, most subjects (79.8%) received antibiotics before specimens' collection, which is an inherent limitation of this observational study due to early antibiotics administration as per national guideline. To overcome this, our inclusion criteria were to enroll subjects within 24 hours of admission, and specimens were collected as soon as possible to hopefully minimize the effects of antibiotics on culture results.”

Lines 279: Mortality numbers are quite high. Please describe more what happened and more specifically the demographics of dead patients.

Thank you for this helpful suggestion. Demographic characteristics of deceased subjects are described in the text. We have also added clinical summary of fatal cases as **Supplementary Table 3**.

Line 296-297 now read:

A clinical summary of the fatal cases is shown in Supplementary Table 3.

Supplementary Table 3. Summary of fatal cases.

Case, Site, Gender (Age, mo)	Medical History	Signs and Symptoms (SS), Vital Signs (VS), Laboratory parameter (Lab) at admission	CXR	Causative Pathogen	ABX during Hospitalization	Hospitalization status	Cause of Death
#01, SMG, Male (4)	Recurrent pneumonia, congenital heart disease, severe malnutrition	 • SS: Cough, fever, dyspnea, chest indrawing, intercostal retraction, rhonchi • VS: 38°C, RR 44x/min, SpO₂ 97% • Lab: Hb 9.6 g/dL, WBC 24.1 ×10⁹/L, PLT 350 ×10⁹/L, NLR 4.63, CRP 25.70 mg/L, PCT 2.41 ng/mL 	Alveolar infiltrate	Rhinovirus, H. influenzae non-type b	Ampicillin, Gentamicin, Ceftriaxon, Cefoperazone Sulbactam	On mechanical ventilator ICU admission (25 days) Died on Day-26	Cardiopulmonary failure Sepsis
#02, SMG, Female (23)	Recurrent pneumonia, congenital heart disease, incomplete NIP (DPT-Hib), malnutrition, development	 • SS: Cough, fever, dyspnea, chest indrawing, intercostal retraction, rhonchi • VS: 37.5°C, RR 56x/min, SpO₂ 95% • Lab: Hb 10.6 g/dL, WBC 14.1 ×10⁹/L, PLT 405 ×10⁹/L, NLR 9.63, CRP 14.90 mg/L, PCT 0.37 ng/mL 	Alveolar and interstitial infiltrates	Influenza A (H1N1)	Ampicillin, Gentamicin, Metronidazole, Ceftriaxon, Meropenem	On mechanical ventilator ICU admission (9 days) Died on Day 21	Cardiopulmonary failure

Case, Site, Gender (Age, mo)	Medical History	Signs and Symptoms (SS), Vital Signs (VS), Laboratory parameter (Lab) at admission	CXR	Causative Pathogen	ABX during Hospitalization	Hospitalization status	Cause of Death
	ental delay						
#03, SMG, Female (11)	Low birth weight, congenital heart disease, incomplete NIP (Measles), severe malnutrition, developmental delay	 • SS: Cough, fever, dyspnea, diarrhea, nasal flaring, chest indrawing, intercostal retraction, rhonchi • VS: 38.3°C, RR 45x/min, SpO₂ 96% • Lab: Hb 8.1 g/dL, WBC 15.9 ×10⁹/L, PLT 677 ×10⁹/L, NLR 1.87 	Alveolar and interstitial infiltrates	Influenza A (H3N2), B. pertussis , H. influenzae non-type b , K. pneumoniae	Ampicillin, Gentamicin, Azithromycin	On nasal cannula Died on day 19	Cardiopulmonary failure
#04, SMG, Male (45)	Recurrent pneumonia, frontometaphyseal dysplasia syndrome, epilepsy, developmental delay	 • SS: Cough, fever, dyspnea, nasal flaring, intercostal retraction, rhonchi, wheezing • VS: 36.7°C, RR 40x/min, SpO₂ 99% • Lab: Hb 13.7 g/dL, WBC 11.3 ×10⁹/L, PLT 277 ×10⁹/L, NLR 0.98, CRP 0.10 mg/L, PCT 0.05 ng/mL 	Alveolar infiltrate	Unknown	Ampicillin, Gentamicin	On Nasal cannula Died on day 2	Respiratory failure
#05, SMG, Male (5)	Premature birth, low birth weight, recurrent pneumonia, congenital heart disease, incomplete NIP (DPT-Hib)	 • SS: Cough, dyspnea, nasal flaring, chest indrawing, intercostal retraction, • VS: 36.8°C, RR 30x/min, SpO₂ 98% • Lab: Hb 10.9 g/dL, WBC 12.4 ×10⁹/L, PLT 396 ×10⁹/L, CRP 0.80 mg/L, PCT 128 ng/mL 	Alveolar infiltrate	K. pneumoniae	Ampicillin, Gentamicin	On Simple mask ICU admission (1 day) Died on day 6	Cardiopulmonary failure

Case, Site, Gender (Age, mo)	Medical History	Signs and Symptoms (SS), Vital Signs (VS), Laboratory parameter (Lab) at admission	CXR	Causative Pathogen	ABX during Hospitalization	Hospitalization status	Cause of Death
#06, SMG, Female (3)	Recurrent pneumonia, incomplete NIP (DPT-Hib), malnutrition	 • SS: Cough, dyspnea, chest indrawing, intercostal retraction, rhonchi • VS: 36.7°C, RR 42x/min, SpO₂ 99% • Lab: Hb 8.2 g/dL, WBC 16 ×10⁹/L, PLT 499 ×10⁹/L, ANC 6.7, NLR 0.76, CRP 13.10 mg/L, PCT 0.28 ng/mL 	Alveolar infiltrate	Unknown	Ampicillin, Gentamicin, Vancomycin, Metronidazole, Meropenem	On mechanical ventilator ICU admission (7 days) Died on day 18	Septic shock, respiratory failure
#07, YGY, Female (10)	Congenital heart disease, incomplete NIP (DPT-Hib, and Measles), severe malnutrition, developmental delay	 • SS: Cough, fever, dyspnea, head bobbing, chest indrawing, intercostal retraction, rhonchi • VS: 39.0 °C, RR 64x/min, SpO₂ 96% • Lab: Hb 10.1 g/dL, WBC 12.1 ×10⁹/L, PLT 415 ×10⁹/L, ANC 6.0, NLR 1.15, CRP 4.90 mg/L, PCT 0.11 ng/mL 	Alveolar infiltrate	hMPV, RSV A	Ampicillin, Gentamicin, Ceftriaxone, Cotrimoxazole	On mechanical ventilator/ ICU admission (13 days) Died on day 17	Sepsis, Pulmonary crisis due to pulmonary hypertension
#08, YGY, Female (3)	Low birth weight, congenital heart disease, incomplete NIP (DPT-Hib), severe malnutrition	 • SS: Cough, fever, dyspnea, chest indrawing, intercostal retraction, rhonchi • VS: 37.2 °C, RR 49x/min, SpO₂ 56% • Lab: Hb 9.7 g/dL, WBC 11.3 ×10⁹/L, PLT 115 ×10⁹/L, ANC 7.0, NLR 1.92 	Alveolar and interstitial infiltrates	Unknown	Ampicillin, Ceftriaxone	On nasal cannula Hospital discharge on day 10 Died on day 29 (outside hospitalization)	Acute Respiratory Distress Syndrome
#09, YGY,	Congenital heart disease, incomplete NIP	 • SS: Cough, dyspnea, inability to drink, nasal flaring, chest indrawing, 	Alveolar and interstitial	H. influenzae non-type b,	Ampicillin, Gentamicin	On nasal cannula	Aspiration, mucous hypersecretion

Case, Site, Gender (Age, mo)	Medical History	Signs and Symptoms (SS), Vital Signs (VS), Laboratory parameter (Lab) at admission	CXR	Causative Pathogen	ABX during Hospitalization	Hospitalization status	Cause of Death
Female (5)	(DPT-Hib), severe malnutrition	intercostal retraction, rhonchi  • VS: 37.0 °C, RR 60x/min, SpO₂ 96% • Lab: Hb 10.3 g/dL, WBC 26.9 ×10⁹/L, PLT 788 ×10⁹/L, ANC 18.5, NLR 2.97 	infiltrates	K. pneumoniae		Died on day 15	
#10, YGY, Male (6)	Recurrent pneumonia, congenital heart disease, tuberculosis, incomplete NIP (DPT-Hib)	 • SS: Cough, fever, dyspnea, nasal flaring, chest indrawing, intercostal retraction, rhonchi, wheezing • VS: 37.3 °C, RR 50x/min, SpO₂ 89% • Lab: Hb 11.6 g/dL, WBC 13.3 ×10⁹/L, PLT 189 ×10⁹/L, ANC 3.7, NLR 0.48, CRP 4.90 mg/L, PCT 0.08 ng/mL 	Alveolar and interstitial infiltrates, pleural effusion	K. pneumoniae	Ampicillin, Gentamicin, Ceftriaxone	On non-rebreather mask Died on day 4	Septic shock
#11, TRG, Female (5)	Premature birth, developmental delay	 • SS: Cough, fever, dyspnea, nasal flaring, rhonchi, wheezing • VS: 37.5 °C, RR 48x/min, SpO₂ 31% • Lab: Hb 8.5 g/dL, WBC 12.1 ×10⁹/L, PLT 208 ×10⁹/L, ANC 8.6, NLR 3.23, CRP 0.91 mg/L, PCT 0.74 ng/mL 	Alveolar infiltrate	A. baumannii (MDR)	Cefotaxime	On Nasal cannula Hospital discharge on day 7 Died on day 17 (outside hospitalization)	Unknown death
#12, TRG, Female (2)	Incomplete NIP (DPT-Hib)	 • SS: Cough, fever, dyspnea, diarrhea, skin rash, intercostal retraction, rhonchi, wheezing 	Alveolar and interstitial infiltrates	Unknown	Ceftriaxone, Ceftazidime, Azithromycin	On Nasal cannula Died on day 8	Sepsis

Case, Site, Gender (Age, mo)	Medical History	Signs and Symptoms (SS), Vital Signs (VS), Laboratory parameter (Lab) at admission	CXR	Causative Pathogen	ABX during Hospitalization	Hospitalization status	Cause of Death
		 • VS: 37.6 °C, RR 63x/min, SpO₂ 93% • Lab: Hb 10.5 g/dL, WBC 13.6 x10⁹/L, PLT 289 x10⁹/L, ANC 10.2, NLR 3.95, CRP 175.30 mg/L, PCT 0.7 ng/mL 					
#13, TRG, Female (2)	Incomplete NIP (DPT-Hib)	 • SS: Cough, fever, dyspnea, nasal flaring, chest indrawing, intercostal retraction, rhonchi • VS: 36 °C, RR 45x/min, SpO₂ 96% • Lab: Hb 7.8 g/dL, WBC 21.2 x10⁹/L, PLT 563 x10⁹/L, ANC 16.5, NLR 3.9, CRP 280.30 mg/L, PCT 0.09 ng/mL 	Alveolar and interstitial infiltrates, pleural effusion	Influenza B, S. mitis (MDR)	Ceftazidime	On non-rebreather mask ICU admission (3 days) Died on day 3	Respiratory Failure
#14, TRG, Female (2)	Congenital heart disease, incomplete NIP (DPT-Hib), severe malnutrition	 • SS: Cough, fever, dyspnea, nasal flaring, chest indrawing, intercostal retraction, rhonchi, wheezing • VS: 37 °C, RR 60x/min, SpO₂ 76% • Lab: Hb 9.5 g/dL, WBC 17.2 x10⁹/L, PLT 296 x10⁹/L, ANC 8.8, NLR 1.42, CRP 0.70 mg/L, PCT 0.02 ng/mL 	Interstitial infiltrate	Unknown	Cefotaxime	On Simple mask Died on day 2	Respiratory Failure
#15, TRG, Male (9)	Incomplete NIP (Measles)	 • SS: Cough, fever, dyspnea, nasal flaring, chest indrawing, intercostal retraction, rhonchi 	Interstitial infiltrate	H. influenzae non-type b	Cefotaxime, Ceftriaxone,	On mechanical ventilator	Meningoencephalitis, Respiratory Failure

Case, Site, Gender (Age, mo)	Medical History	Signs and Symptoms (SS), Vital Signs (VS), Laboratory parameter (Lab) at admission	CXR	Causative Pathogen	ABX during Hospitalization	Hospitalization status	Cause of Death
		 • VS: 37 °C, RR 30x/min, SpO₂ 89% • Lab: Hb 6.4 g/dL, WBC 25.7 ×10⁹/L, PLT 801 ×10⁹/L, ANC 18.5, NLR 3.43, CRP 33.35 mg/L, PCT 0.34 ng/mL 			Meropenem	ICU admission (8 days) Died on day 12	
#16, TRG, Female (4)	Premature birth, low birth weight, congenital heart disease, incomplete NIP (DPT-Hib)	 • SS: Cough, fever, dyspnea, diarrhea, chest indrawing, intercostal retraction, rhonchi • VS: 38 °C, RR 32x/min, SpO₂ 85% • Lab: Hb 9.2 g/dL, WBC 16.8 ×10⁹/L, PLT 224 ×10⁹/L, ANC 9.4, NLR 2.24, CRP 2.46 mg/L, PCT 2.24 ng/mL 	Alveolar and interstitial infiltrates,	H. influenzae non-type b, K. pneumoniae	Cefotaxime, Gentamicin, Ceftriaxone	On nasal cannula Died on day 11	Unknown death
#17, TRG, Female (20)	Developmental delay, incomplete NIP (DPT-Hib)	 • SS: Cough, fever, dyspnea, chest indrawing, intercostal retraction, rhonchi • VS: 36.3°C, RR 40x/min, SpO₂ 75% • Lab: Hb 7.0 g/dL, WBC 15.2 ×10⁹/L, PLT 668 ×10⁹/L, ANC 9.7, NLR 2.13, CRP 55.10 mg/L 	Alveolar and interstitial infiltrates, pleural effusion	H. influenzae non-type b, K. pneumoniae	Cefotaxime, Gentamicin, Ceftriaxone	On mechanical ventilator ICU admission (3 days) Died on day 8	Septic shock, Cardiopulmonary failure
#18, TRG, Male (4)	Low birth weight, developmental delay, recurrent pneumonia, incomplete NIP	 • SS: Cough, fever, dyspnea, nasal flaring, chest indrawing, intercostal retraction, rhonchi • VS: 36.7 °C, RR 30x/min, SpO₂ 92% • Lab: Hb 11.6 g/dL, WBC 20.5 ×10⁹/L, PLT 433 	Alveolar and interstitial infiltrates,	PIV 3, H. influenzae non-type b, S. pneumoniae	Ceftazidime	On non-rebreather mask Died on day 3	Respiratory failure

Case, Site, Gender (Age, mo)	Medical History	Signs and Symptoms (SS), Vital Signs (VS), Laboratory parameter (Lab) at admission	CXR	Causative Pathogen	ABX during Hospitalization	Hospitalization status	Cause of Death
	(DPT-Hib), severe malnutrition	$\times 10^9/L$, ANC 11.9, NLR 2.52, CRP 16.80 mg/L, PCT 20.1 ng/mL					
#19, TRG, Male (15)	Incomplete NIP (DPT-Hib and Measles)	 • SS: Cough, fever, dyspnea, rhonchi • VS: 37.8 °C, RR 52x/min, SpO₂ 80% • Lab: Hb 9.4 g/dL, WBC 23.6 $\times 10^9/L$, PLT 786 $\times 10^9/L$, CRP 3.30 mg/L, PCT 0.07 ng/mL 	Interstitial infiltrate	RSV, B, B, pertussis, H. influenzae non-type b	Cefotaxime	On nasal cannula Hospital discharged on day 5 Died on day 20 (outside hospitalization)	Unknown death

Abbreviation. SMG: Semarang site; YGY: Yogyakarta site; TGR: Tangerang site; NIP: mandatory National Immunization Program; DPT-Hib: a combined vaccine of adsorbed diphtheria, tetanus toxoids, acellular pertussis and of *Haemophilus influenzae* type b conjugate vaccines; CXR: chest X-ray; ABX: Antibiotics; RSV: Respiratory Syncytial Virus; hMPV: Human Metapneumovirus; PIV: Parainfluenza Virus; MDR: Multiple drug resistance.

Lines 377-378 are the most important sentences of this study and I would have expected to solve this issue in a prospective study. If cultures/PCR results do not impact care, what is specifically its clinical and practical usefulness??

Thank you for highlighting this important point. We apologize for the confusion. While molecular assays have limitations (as stated in **lines 388-391**), they are very useful for identification of specific pathogens. Pathogen identification facilitates targeted clinical management, which limits unnecessary patient risks and facilitates antimicrobial stewardship. We found that PCR had greater sensitivity for identification of bacterial pathogens than blood or sputum culture. Capacity for molecular assays is quite limited in Indonesia but has improved during COVID-19 pandemic. We recommend incorporation of molecular assays into standard of care for children with CAP. We also mentioned on the text that conventional culture methods will still be required to confirm phenotypic resistance in clinical setting (**lines 380-382**).

We have edited the conclusions for clarification. **Lines 418-422** now read:

“PCR on IS demonstrated the best sensitivity for pathogen identification. We recommend incorporating molecular assays for pathogen detection, preferably multiplexed point-of-care assays, into practice guidelines. Improvements in Indonesia’s lab infrastructure during the COVID-19 pandemic can be leveraged to facilitate use of molecular assays for evaluation of childhood CAP.”

Lines 397-402: how can you be sure that detected pathogens are causative pathogens??

Thank you for raising this important philosophical issue. It is true that we can't be 100% certain a detected pathogen is causative. Given the challenge of determining whether a pathogen is actually causing a clinical picture, we developed an algorithm based on available evidence and expert input. The algorithm has been previously published (<https://doi.org/10.3389/fped.2021.629318>) and is cited on **lines 178-189**.

A huge number of patients received antibiotics before culture/swab >> this needs to be seriously discussed

Thank you for this comment. Similar to our response above, we have modified some text in the current manuscript to discuss antibiotic administration prior to culture/swab.

Lines 399-405 now read:

"The relatively small sample size, geographic limitation to the island of Java and observational design may limit generalizability and causal inference. Second, most subjects (79.8%) received antibiotics before specimens' collection, which is an inherent limitation of this observational study due to early antibiotics administration as per national guideline. To overcome this, our inclusion criteria were to enroll subjects within 24 hours of admission, and specimens were collected as soon as possible to hopefully minimize the effects of antibiotics on culture results."

Lines 400-401 state: "several subjects had pneumonia of unknown etiology; this may have been due to administration of antibiotics before culture which could reduce sensitivity

Risks of unnecessary antibiotics are also further emphasized in **lines 329-334**:

"In PEER-PePPeS, viruses were found in 57% of subjects (virus only + mixed infection), with 16.5% of cases attributed to virus only. Thus, many patients probably received unnecessary antibiotics when they were covered empirically according to current Indonesia guidelines. Improving ability to discriminate between viral and bacterial infections would facilitate optimization of antibiotic administration and counter antimicrobial resistance, a major global health challenge.[31]"

Impact on clinical care and guidelines derived from study findings should be outlined in the conclusions. this is the message we are waiting for.

Thank you for the suggestion. **Lines 418-423** now read:

"PCR on IS demonstrated the best sensitivity for pathogen identification. We recommend incorporating molecular assays for pathogen detection, preferably multiplexed point-of-care assays, into practice guidelines. Improvements in Indonesia's lab infrastructure during the COVID-19 pandemic can be leveraged to facilitate use of molecular assays for evaluation of childhood CAP. Optimization of pathogen detection to understand changing childhood CAP epidemiology will also inform public policy on prevention and management."

Add this paper to the references:

Tannous R, Haddad RN, Torbey PH. Management of Community-Acquired Pneumonia in Pediatrics: Adherence to Clinical Guidelines. *Front Pediatr.* 2020;8:302. Published 2020 Jun 19. doi:10.3389/fped.2020.00302

Thank you, the article has provided us with good insight. The paper is now cited as reference article #10.

Reviewer: 2

Dr. Tu-Hsuan Chang, Chi Mei Medical Center

Competing interests of Reviewer: I have no competing interests.

Comments to the Author:

Thank you for inviting me to review this manuscript. This is a prospective study using comprehensive approach to investigate common CAP pathogens in Indonesia children. The article is well written and the method is rigorous. It provided valuable epidemiological data. I really appreciate the work done by all investigators. I have just a few comments and suggestions.

We would like to thank the reviewer for the positive and constructive comments. We have carefully considered the reviewer's comments and provide itemized responses below.

1. To support the description of "subject characteristics were similar across the three study sites", related statistical analysis should be presented in table 1.

Thank you for this helpful suggestion. Statistical analysis in the methods section has been modified.

Lines 195-198 now read:

"Differences in categorical variables were compared using Pearson χ^2 or Fisher's exact test when the expected values in any of the contingency table cells were below 5. Differences in continuous variables were compared using One-way ANOVA or Kruskal-Wallis H test for data which did not follow the normal distribution based on Levene's test."

We also modified the table 1 to include p -values of comparisons across sites and revised some sentences in the text based on statistically significant results. In specific, **Lines 216-217**: The statement, "subject characteristics were similar across the three study sites" has been modified to, "Age, gender, laboratory values, and pneumonia severity by WHO classification were similar across the three study sites."

Lines 216-223 now read:

"Demographic and clinical characteristics are presented in **Table 1**. Age, gender, laboratory values, and pneumonia severity by WHO classification were similar across the three study sites. The median age was nine months (IQR, 5 to 20), and 54.7% of subjects were male. The most common comorbid conditions were developmental delay (27.7%), congenital heart disease (26.1%), and severe malnutrition (18.6%), with subjects from Yogyakarta site having the greatest proportion of those comorbidities. The percentage of subjects who had been vaccinated (age-adjusted) against pneumococcus, influenza, Hib-DPT, and measles vaccines were 2.1%, 1.1%, 55.9%, and 75.0%, respectively."

Table 1. Baseline Characteristics of Subjects.

Demographic Characteristics	All (N=188)	Semarang (N=47)	Yogyakarta (N=52)	Tangerang (N=89)	P-value
Age, median (IQR) months	9 (5 – 20)	9 (5.5 – 21)	8 (4 – 13.3)	11 (5-20)	0.442
Gender, Male, (%)	103 (54.7)	29 (61.7)	26 (50)	48 (53.9)	0.493
Household Characteristics, (%):	163 (86.7)	37 (78.7)	43 (82.7)	84 (94.3)	0.019
• Low Education of Parents*	121 (64.4)	19 (40.4)	42 (80.8)	60 (67.4)	<0.001
• Living in a dense neighborhood†	70 (37.2)	12 (25.5)	29 (55.8)	29 (32.6)	0.004
• Living near waste disposal	109 (58.0)	22 (46.8)	43 (82.7)	44 (49.4)	<0.001
• Sick household contact <14 days	120 (63.8)	24 (51.1)	27 (51.9)	69 (77.5)	0.001
• Exposure to cigarette smoke	4 (2.1)	2 (4.3)	1 (1.9)	1 (1.1)	0.374
• Attending daycare					
Medical history (%):					
• Premature baby	34 (18.1)	4 (8.5)	16 (30.8)	14 (15.7)	0.012
• Low birth weight	46 (24.4)	12 (25.5)	20 (38.5)	14 (15.7)	0.011
• Developmental delay	52 (27.7)	16 (34.0)	21 (40.4)	15 (16.8)	0.003
• Congenital heart disease	49 (26.1)	16 (34.0)	24 (46.2)	9 (10.1)	<0.001
• Severe malnutrition‡	35 (18.6)	10 (21.3)	13 (25.0)	12 (13.5)	0.205
• Neurological disorder	25 (13.3)	5 (10.6)	17 (32.7)	3 (3.4)	<0.001
• Asthma	9 (4.8)	3 (6.4)	1 (1.9)	5 (5.6)	0.563
• HIV disease§	2 (1.1)	1 (2.1)	1 (1.9)	0 (0)	0.315
• Tuberculosis (recent/cured)	10 (5.3)	4 (8.5)	2 (3.8)	4 (4.5)	0.588
Immunization history, fully vaccinated for age (%):					
• DPT-Hib	105 (55.9)	30 (63.8)	25 (48.1)	50 (56.2)	0.233
• Influenza	2 (1.1)	0 (0)	2 (3.8)	0 (0)	0.132
• Pneumococcus	4 (2.1)	0 (0)	4 (7.7)	0 (0)	0.009
• Measles	141 (75.0)	38 (80.9)	41 (78.8)	62 (69.7)	0.175

Demographic Characteristics	All (N=188)	Semarang (N=47)	Yogyakarta (N=52)	Tangerang (N=89)	P-value
Symptoms and signs (%):					
• Cough	171 (91.0)	40 (85.1)	42 (80.8)	89 (100)	<0.001
• Shortness of breath	174 (92.6)	41 (87.2)	48 (92.3)	85 (95.5)	0.214
• Fever	152 (80.9)	34 (72.3)	35 (67.3)	83 (93.3)	<0.001
• Decreased Consciousness	7 (3.7)	1 (2.1)	1 (1.9)	5 (5.6)	0.612
• Inability to drink	13 (6.9)	4 (8.5)	5 (9.6)	4 (4.5)	0.425
• Diarrhea	36 (19.1)	6 (12.8)	4 (7.7)	26 (29.2)	0.003
• Vomiting	14 (7.4)	4 (8.5)	5 (9.6)	5 (5.6)	0.595
• Seizure	6 (3.2)	1 (2.1)	0 (0)	5 (5.6)	0.203
• Fast breathing	80 (42.6)	15 (31.9)	43 (82.7)	22 (24.7)	<0.001
• Intercostal retraction	171 (91.0)	43 (91.5)	52 (100)	76 (85.4)	0.005
• Rhonchi	168 (89.4)	42 (89.4)	39 (75.0)	87 (97.8)	<0.001
• Wheezing	35 (18.6)	9 (19.1)	10 (19.2)	16 (18.0)	1.000
• Chest indrawing	125 (66.5)	36 (76.6)	43 (82.7)	46 (51.7)	<0.001
• SpO ₂ <90% and/or Cyanosis	43 (22.9)	7 (14.9)	17 (32.7)	19 (21.3)	0.098
Leukocyte count, median (IQR) × 10 ³ /uL	14.0 (10.4 – 18.9)	14.9 (11.1 – 18.8)	12.1 (9.8 – 17.8)	14.0 (10.4 – 19.0)	0.356
Neutrophil-lymphocyte ratio (NLR), median (IQR)	1.4 (0.9 – 2.8)	1.3 (0.9 – 2.6)	1.0 (0.6 – 2.0)	1.9 (1.1 – 3.2)	0.367
CRP, median (IQR) mg/L	9.0 (3.6 – 28.0)	11.8 (1.6 – 23.3)	9.0 (4.9 – 21.8)	8.4 (1.5 – 34.1)	0.665
PCT, median (IQR) ng/mL	0.2 (0.1 – 1.7)	0.2 (0.1 – 1.5)	0.2 (0.1 – 1.0)	0.2 (0.1 – 2.6)	0.912
Severe pneumonia (WHO Classification 2014 version) (%)	89 (47.3)	26 (55.3)	26 (50.0)	37 (41.6)	0.281
CXR Findings (%):					
• Pleural effusion	5 (2.7)	1 (2.1)	2 (3.8)	2 (2.2)	0.850
• Interstitial infiltrate	131 (69.7)	26 (55.3)	30 (57.7)	75 (84.3)	<0.001
• Alveolar infiltrate	125 (66.5)	41 (87.2)	44 (84.6)	40 (44.9)	<0.001

Demographic Characteristics	All (N=188)	Semarang (N=47)	Yogyakarta (N=52)	Tangerang (N=89)	P-value
Antibiotic administration prior to blood collection for blood culture (%):	150 (79.8)	39 (83.0)	49 (94.2)	62 (69.7)	0.002
• Ampicillin					
• Ampicillin – Gentamicin	14 (7.4)	5 (10.6)	9 (17.3)	0 (0)	
• Cefotaxime	64 (34.0)	25 (53.2)	39 (75.0)	0 (0)	
• Ceftriaxone	33 (17.5)	0 (0)	0 (0)	33 (37.1)	
• Others [¶]	26 (13.8)	0 (0)	0 (0)	26 (29.2)	
	13 (6.9)	9 (19.1)	1 (1.9)	3 (3.4)	

^{*}Low education of parents was defined by highest level of parents' formal education being high school diploma or less; [†]A densely populated neighborhood was defined as >200 people/km² or <8 m²/person in the subject's home; [‡] Severe malnutrition was defined as weight for height below -3 standard deviations from the median of the WHO Child Growth Standards; [§]Subjects were tested for HIV infection if a parent / guardian provided consent and a specimen was available (n=160); ^{||}Full vaccination was defined as being up to date for age per vaccination schedule at study enrollment; [¶]Other antibiotics include: Amoxicillin, Cefamandole, Ceftazidime, Gentamicin, Cefotaxime – Amikacin, Ceftriaxone – Gentamicin, and Azithromycin – Ampicillin – Gentamicin

2. The first letter of virus should be a lower case

Thank you for the correction. We have changed the relevant instances of “virus” in the manuscript.

3. Some abbreviations (e.g.: “IS” specimens in abstract) need to be spell out when first appeared.

Thank you for your suggestion. **Lines 53-54** in the abstract now read:

“The polymerase chain reaction (PCR) assays on induced sputum (IS) specimens captured most of the pathogens identified in this study.”

NP and IS are also defined in the main text on **line 149**:

“nasopharyngeal (NP) swab for molecular tests; induced sputum (IS) for culture and molecular tests”

VERSION 2 – REVIEW

REVIEWER	N. Haddad, Raymond Necker-Children's Institute, Pediatric and Congenital Cardiology
REVIEW RETURNED	30-Apr-2022

GENERAL COMMENTS	I would like to thank the authors for the efforts they made to revise their manuscript. This observational study could be of interest to physicians living in the nearby regions where the study was conducted. It comprehensively and extensively describes the
--

	reality of CAP in Indonesian Regions which I find a positive step to improving local care. Overall, I still find the discussion a bit too long, thereby suggesting shortening it a little bit could be helpful. A review of the paper by a native English speaker could be also helpful to enhance the readability of some sentences. The authors state that a total of 150 (79.8%) children received antibiotics prior to the collection of blood for culture which is a relatively high proportion. Since the authors had no action on this inappropriate prescription of antibiotics, It would be interesting to detail in the results (if data is available) what they received and how many doses during the 24 hours? I would recommend adding to Fig.1 chart summarising the proportion of children receiving each combination of different diagnostic tests. Lines 257-258: please add % as previously requested. Same in the mortality paragraph In table 2, replace 0(0%) by --
--	--

REVIEWER	Chang, Tu-Hsuan Chi Mei Medical Center, Department of Pediatrics
REVIEW RETURNED	14-Apr-2022

GENERAL COMMENTS	The authors have addressed all the issues or concerns. I have no further comment on this manuscript.
--

VERSION 2 – AUTHOR RESPONSE

Reviewer Reports:

Reviewer: 1

Dr. Raymond N. Haddad, Necker-Children's Institute

Comments to the Author:

I would like to thank the authors for the efforts they made to revise their manuscript. This observational study could be of interest to physicians living in the nearby regions where the study was conducted. It comprehensively and extensively describes the reality of CAP in Indonesian Regions which I find a positive step to improving local care.

We thank the reviewer for re-reviewing our manuscript. We have carefully considered the reviewer's helpful comments. Our responses are below.

Overall, I still find the discussion a bit too long, thereby suggesting shortening it a little bit could be helpful. A review of the paper by a native English speaker could be also helpful to enhance the readability of some sentences.

Thank you for this comment. The discussion section has been revised for clarity and brevity. The native-speaker authors edited and approved this final manuscript.

The authors state that a total of 150 (79.8%) children received antibiotics prior to the collection of blood for culture which is a relatively high proportion. Since the authors had no action on this inappropriate prescription of antibiotics, It would be interesting to detail in the results (if data is available) what they received and how many doses during the 24 hours?

Thank you for your input. Additional information about antibiotic administration has been added to the main manuscript. Details including frequency of specific regimens and dosing are listed in the new Supplementary Table 2. Numbering of Supplementary Tables 3 and 4 has been adjusted accordingly.

Lines 237-241 now read:

All 188 enrolled cases were treated with antibiotics, and 150 of them (79.8%) had received 1 to 2 doses of antibiotics prior to collection of blood culture in the emergency unit, with the combination of ampicillin and gentamicin (34.6%), cefotaxime (17.0%), and ceftriaxone (14.4%) being the three most frequent regimens used. Details of antibiotic regimens administered before blood culture, including dosage and given frequency, are presented in **Supplementary Table 2**.

Supplementary table 2 now reads:

Supplementary Table 2. Antibiotic regimens administered prior to blood culture

Antibiotic Regimen, (Dose)	All sites (N=188), Administered Dose(s) prior to blood culture, N (%)	Semarang (N=47) Administered Dose(s) prior to blood culture, N (%)	Yogyakarta (N=52) Administered Dose(s) prior to blood culture, N (%)	Tangerang (N=89) Administered Dose(s) prior to blood culture, N (%)
Ampicillin (50 mg/kg IV q6hr) + Gentamicin (2 – 7.5 mg/kg IV q24hr)	65 (34.6) 1x: 45 (24.0) 2x: 20 (10.6)	25 (53.2) 1x: 20 (42.6) 2x: 5 (10.6)	40 (76.9) 1x: 25 (48.1) 2x: 15 (28.8)	0 (0)

Antibiotic Regimen, (Dose)	All sites (N=188), Administered Dose(s) prior to blood culture, N (%)	Semarang (N=47) Administered Dose(s) prior to blood culture, N (%)	Yogyakarta (N=52) Administered Dose(s) prior to blood culture, N (%)	Tangerang (N=89) Administered Dose(s) prior to blood culture, N (%)
Cefotaxime (50 – 100 mg/kg IV q6hr)	32 (17.0) All received 1 dose	0 (0)	0 (0)	32 (36.0) All received 1 dose
Ceftriaxone (50 mg/kg IV q12hr)	27 (14.4) All received 1 dose	0 (0)	0 (0)	27 (30.3) All received 1 dose
Ampicillin (50 mg/kg IV q6hr)	14 (7.4) 1x: 10 (5.3) 2x: 4 (2.1)	5 (10.6) All received 1 dose	9 (17.3) 1x: 5 (9.6) 2x: 4 (7.7)	0 (0)
Gentamicin (2 – 7.5 mg/kg IV q24hr)	3 (1.6) 1x: 2 (1.1) 2x: 1 (0.5)	3 (6.4) 1x: 2 (4.3) 2x: 1 (2.1)	0 (0)	0 (0)
Ceftazidime (50 – 100 mg/kg IV q8hr)	3 (1.6) All received 1 dose	0 (0)	0 (0)	3 (3.4) All received 1 dose
Cefamandole (50 – 100 mg/kg IV q12hr)	2 (1.1) 1x: 1 (0.5) 2x: 1 (0.5)	2 (4.3) 1x: 1 (2.1) 2x: 1 (2.1)	0 (0)	0 (0)
Ceftriaxone (50 mg/kg IV q12hr) + Gentamicin (2 – 7.5 mg/kg IV q24hr)	2 (1.1) All received 1 dose	2 (4.3) All received 1 dose	0 (0)	0 (0)

Antibiotic Regimen, (Dose)	All sites (N=188), Administered Dose(s) prior to blood culture, N (%)	Semarang (N=47) Administered Dose(s) prior to blood culture, N (%)	Yogyakarta (N=52) Administered Dose(s) prior to blood culture, N (%)	Tangerang (N=89) Administered Dose(s) prior to blood culture, N (%)
Amikacin (15 mg/kg IV q8hr) + Cefotaxime (50 – 100 mg/kg IV q6hr)	1 (0.5) All received 1 dose	1 (2.1) All received 1 dose	0 (0)	0 (0)
Amoxicillin syrup (40 mg/kg PO q12hr)	1 (0.5) All received 1 dose	1 (2.1) All received 1 dose	0 (0)	0 (0)

IV = intravenous; PO = peroral; qXhr = given at X hour intervals.

I would recommend adding to Fig.1 chart summarising the proportion of children receiving each combination of different diagnostic tests.

Thank you for your input. We have modified Figure 1 as follows.

Figure 1. Subject screening, enrolment, and monitoring flowchart. CAP, community-acquired pneumonia; RR, respiratory rate; CXR, chest X-Ray; CRP, C-reactive protein; PCT, procalcitonin; NP, nasopharyngeal; IS, induced sputum; WB, whole blood; BC, blood culture; IS, induced sputum culture; PCR, polymerase chain reaction.

Lines 257-258: please add % as previously requested.

Same in the mortality paragraph

Thank you for your reminder. We apologize for the oversight. We now have added the % as originally requested.

Lines 256 – 260 now read:

H. influenzae non-type B (N=73, 38.8%), RSV (N=51, 27.1%), *K. pneumoniae* (N=43, 22.9%), *S. pneumoniae* (N=29, 15.4%), Influenza virus (N=25, 13.3%), *S. aureus* (N=20, 10.6%), PIV (N=17, 9.0%), hMPV (N=11, 5.8%), Rhinovirus (N=10, 5.3%), and *B. pertussis* (N=7, 3.7%) were the top ten pathogens identified, more commonly appearing in mixed infection as opposed to as a sole pathogen (Fig 2. Panel C).

Lines 284 – 298 now read:

Nineteen (10.1%) of the 188 subjects died during the 30-day study period. Seven (36.8%) of these 19 were male, and most (N=17, 89.5%) were less than 1 year old. Among the 19 deceased subjects, median study duration was 12 (IQR, 4 – 17.5) days; eight (42.1%) were admitted to ICU, and six (31.6%) received mechanical ventilation. Twelve (63.2%) died due to respiratory failure, three (15.8%) due to sepsis, and three (15.8%) for unknown reasons after discharge (data not shown). Most deaths occurred in the 2-11 mo age group compared with the 12-59 mo age group (78.9% vs. 21.1%, $p=0.036$). Causative pathogens for deceased subjects were bacterial-only in seven (36.8%), viral-only in two (10.5%), mixed in five (26.3%), and unknown in five subjects (26.3%). There were no significant differences in pathogen distribution between subjects that survived and died. *H. influenzae* non-type B was the most common pathogen identified in deceased subjects (N=8, with the case fatality rate [CFR] in this study of 11.0%), followed by *K. pneumoniae* (N=6, CFR of 13.9%), Influenza virus (N=3, CFR of 12.0%), *B. pertussis* (N=2, CFR of 28.6%), and RSV (N=2, CFR of 3.9%) (**Supplementary Table 3**). Pre-existing conditions amongst deceased subjects included congenital heart disease (N=10, 52.6%), severe malnutrition (N=7, 36.8%), and developmental delay (N=7, 36.8%). A clinical summary of the fatal cases is shown in **Supplementary Table 4**.

In table 2, replace 0(0%) by --

Thank you. Table 2 now reads:

Table 2. Causative Pathogens per PEER-PePPeS Rules by Detection Method

Pathogen	N	Blood culture N (%)	IS culture N (%)	Whole blood PCR N (%)	NP / OP PCR N (%)	IS PCR N (%)	Serology Test N (%)
Gram-positive cocci bacteria							
S. pneumoniae	29	1 (3.4%)	3 (10.3%)	--	21 (72.4%)	28 (96.6%)	
S. aureus	20	--	7 (35%)	--	11 (55%)	19 (95%)	
S. mitis	4	--	4 (100%)				
S. pyogenes	1	--	1 (100%)				
Gram-negative cocci bacteria							
M. catarrhalis	2	--	2 (100%)		2 (100%)	2 (100%)	
Gram-negative rods bacteria							
H. inf non-type b	73	--	--	8 (10.9%)	60 (82.2%)	71 (98.6%)	
K. pneumoniae	43	--	17 (39.5%)		2 (4.7%)	34 (79.1%)	
B. pertussis	7	--	--			7 (100%)	
E. coli	5	1 (20%)	4 (80%)				
P. aeruginosa	4	--	4 (100%)				
A. baumannii	3	--	3 (100%)				
H. inf type b	2	--	--	--	--	2 (100%)	
N. meningitidis	1	1 (100%)	1 (100%)				
Atypical-bacteria							
C. pneumoniae	5	--	--			--	5 (100%)
M. pneumoniae	5	--	--			5 (100%)	1 (20%)
L. pneumophila	1	--	--			--	1 (100%)
Virus							

RSV	51				36 (70.6%)	45 (88.2%)	10 (19.6%)
RSV A	15				10 (66.7%)	13 (86.7%)	
RSV B	36				26 (72.2%)	32 (88.8%)	
Influenza virus	25				16 (64%)	22 (88%)	9 (36%)
inf A (H1N1)	7				7 (100%)	7 (100%)	7 (70%)
inf A (H3N2)	3				3 (100%)	3 (100%)	
inf B	14				6 (42.9%)	12 (85.7%)	2 (14.3%)
PIV	17				16 (94.1%)	15 (88.2%)	3 (17.6%)
PIV 1	5				5 (100%)	4 (80%)	3 (17.6%)
PIV 2	0				--	--	
PIV 3	11				10 (90.9%)	10 (90.9%)	
PIV 4	1				1 (100%)	1 (100%)	
hMPV	11				5 (45.5%)	10 (90.9%)	
Rhinovirus	10				10 (100%)	6 (60%)	4 (40%)
Enterovirus	5				3 (60%)	3 (60%)	3 (60%)
Bocavirus	3				2 (66.7%)	3 (100%)	
hCoV-NL63	2				2 (100%)	2 (100%)	

Grey-box indicates the assay was not performed

Reviewer: 2

Dr. Tu-Hsuan Chang, Chi Mei Medical Center

Comments to the Author:

The authors have addressed all the issues or concerns. I have no further comment on this manuscript.

Thank you very much for your review.